# CONFLICT-AWARE REPRESENTATION EDITING FOR ROBUST RETRIEVAL-AUGMENTED GENERATION

## ABSTRACT

Large Language Models (LLMs) may not always provide accurate responses to user queries, owing to the staleness of training data and the presence of noise. To address this, Retrieval-Augmented Generation (RAG) has been widely adopted, enabling LLMs to ground their responses in external knowledge sources. Nonetheless, recent studies show that conflicts between the retrieved external knowledge and the model's parametric knowledge can lead to hallucinatory outputs, and this problem is exacerbated when the retrieved documents contain noise. In this work, we propose **Conflict-Aware Representation Editing** (CARE), a method designed to generate robust responses even when the retrieval includes documents with low relevance to the query. CARE aims to produce conflict-resilient responses by editing the internal representations of the model. Assuming that LLMs encode distinguishable internal patterns indicative of knowledge conflicts, we introduce an autoencoder into the model's internal layers to identify such regions. We then modulate neuron activations accordingly, steering the model to generate responses unaffected by knowledge conflicts. We evaluate CARE across six Question Answering (QA) benchmarks and four LLMs, demonstrating its superior performance over existing methods.

## 1 INTRODUCTION

Recent advances in Large Language Models (LLMs) have demonstrated their remarkable capabilities across a wide range of natural language processing tasks, driven by their ability to internalize extensive world knowledge through pretraining on large-scale textual corpora (Alabdulmohsin et al., 2022; Haviv et al., 2023; OpenAI, 2023; Zhao et al., 2025a). Despite their impressive performance, LLMs inherently suffer from limitations in reflecting up-to-date information beyond their static training cut-off, and are susceptible to errors and biases present in the training data (Pagnoni et al., 2021; Luu et al., 2022; Ji et al., 2023; Shi et al., 2024b). These challenges highlight the fundamental difficulty of fully encoding all knowledge within parametric models and underscore the necessity for real-time access to external information.

To address the limitations, Retrieval-Augmented Generation (RAG) has emerged as a widely adopted paradigm, wherein relevant external knowledge is retrieved and incorporated into the model's response generation process (Adelani et al., 2021; Izacard et al., 2023; Rubin & Berant, 2024). While RAG offers a promising solution for enhancing the recency and reliability of generated content, it still faces two fundamental challenges: (1) *__Knowledge Conflict__*: Retrieved documents may contradict the model's parametric knowledge, leading to inconsistencies in the generated response. Such conflicts are known to induce hallucinations, wherein the model produces factually incorrect content (Tan et al., 2024; Xie et al., 2024). (2) *__Retrieval Noise__*: inherent limitations of retrieval systems, retrieved passages may be irrelevant to the query. This not only hinders the model's ability to accurately assess relevance, but also introduces interference when integrating retrieved content with internal knowledge (Dai et al., 2024; Su et al., 2025).

In complex scenarios, a sophisticated mechanism is required to coordinate the relationship between the LLM's internally stored knowledge and the information retrieved from external sources. Existing approaches that focus on the retrieval stage alone do not comprehensively address conflicts arising from interactions with the model (Chan et al., 2024; Kim et al., 2024; Wei et al., 2025). Moreover, methods that adaptively toggle the use of RAG or adjust responses through contrastive decoding

Figure 1: The overall pipeline of CARE. Retrieved context and the question are fed into an LLM to form an internal representation, and an autoencoder encodes it into a latent vector. The model is trained to distinguish latent representations associated with correct versus incorrect answers during training. At inference time, the learned direction vector is applied to the latent representation of a new input to edit the representation, and the modified vector is then injected back into the LLM to steer it toward a more accurate final output.

have limited effectiveness in resolving conflicts intrinsic to the LLM itself (Asai et al., 2023; Jin et al., 2024; Jeong et al., 2024).

To address this problem, we aim to build a robust model under noisy knowledge conflict situations by modifying the latent representations of LLMs. Recent studies report that in the vast high-dimensional representation space of LLMs, different semantic features are arranged in nearly orthogonal, disentangled linear subspaces (Bolukbasi et al., 2016; Ortiz-Jimenez et al., 2023; Jiang et al., 2024; Arditi et al., 2024). Building on this insight, we hypothesize that when external information conflicts with a model's internal knowledge, the internal representations of the model exhibit distinct neuron activation patterns: one reflecting accurate knowledge in a *knowledge-alignment space* and another reflecting misleading or conflicting information in a *knowledge-conflict space*. Based on this, we propose a novel method called **Conflict-Aware Representation Editing** (CARE), which identifies and regulates points of conflict between parametric knowledge and retrieved document-based knowledge within the LLM's internal representation space. We use an autoencoder to operationalize this idea by learning a compact latent space that disentangles knowledge-alignment and knowledge-conflict components in the hidden representations. As illustrated in Figure 1, we train an autoencoder-based module to capture such patterns from neuron activations generated when a query and retrieved documents are jointly provided, thereby identifying conflict regions at intermediate layers. During inference, the trained autoencoder is integrated into specific LLM layers to adjust representations by suppressing conflict signals. This approach effectively mitigates conflicts that arise during inference, ultimately enhancing the reliability of LLM responses even in uncertain situations.

To validate the effectiveness of our approach, we perform comprehensive evaluations on four instruction-tuned LLMs: LLaMA-2-7B-Chat, LLaMA-2-13B-Chat (Touvron et al., 2023), LLaMA-3.1-8B-Instruct (Llama Team, 2024), and Qwen-2.5-7B-Instruct (Qwen et al., 2025), across four Question Answering (QA) benchmarks and two multi-hop QA datasets. CARE achieves superior QA accuracy compared to a wide range of baseline approaches, including RAG that accounts for noisy documents, knowledge mitigation that maximizes the impact of external knowledge, and truthfulness editing during inference, thereby demonstrating robustness in RAG-based settings.

To sum up, our main contributions are as follows:

- We propose a novel internal representation editing method to mitigate conflicts between the parametric knowledge of LLMs and retrieved external information.

- By learning neuron activation patterns with an autoencoder and injecting an intervention module into specific intermediate layers at inference time, CARE aligns neuron activations between internal and external knowledge, ensuring reliable information is properly reflected in the model's response.

- Experimental results across various instruction-tuned LLMs and diverse QA benchmarks demonstrate that `CARE` effectively reduces confusion and incorrect responses caused by internal knowledge conflicts and noisy retrieved documents, outperforming existing baselines.

## 2 RELATED WORK

**Knowledge Conflict**  LLMs can experience conflicts between their parametric knowledge and retrieved external information, which may lead to incorrect responses (Longpre et al., 2021; Neeman et al., 2023; Xu et al., 2024; Shi et al., 2024c). To address knowledge conflicts, various approaches have been proposed. Neeman et al. (2023) introduces a training method that explicitly separates answers based on parametric and contextual knowledge. Wang et al. (2023) proposes a novel instruction-based approach to distinguish between evidence originating from differing perspectives. Zhang & Choi (2023) focuses on identifying and removing outdated facts within LLMs to ensure compliance with up-to-date contextual information. CAD (Shi et al., 2024c) enhances the effect of relevant context by amplifying the difference in output probabilities with and without context. While effective, these approaches must also account for realistic scenarios where the retrieved information may be noisy or unreliable (Du et al., 2022; Xu et al., 2024).

**Retrieval-Augmented Generation**  RAG aims to enhance factual accuracy and reflect up-to-date information by integrating external document-based knowledge into language models (Adelani et al., 2021; Izacard et al., 2023; Rubin & Berant, 2024). Self-RAG (Asai et al., 2023) allows an LLM to retrieve external knowledge only when necessary, and to reflect on both its generation and the retrieved results in order to improve factual accuracy and quality. Adaptive-RAG (Jeong et al., 2024) dynamically toggles the use of retrieval depending on question difficulty, aiming to optimize response quality. SURE (Kim et al., 2024) improves consistency by summarizing retrieved documents and comparing them with candidate answers to assess supporting evidence. While these methods focus on improving input-level consistency, they do not fully resolve conflicts with the model's internal knowledge. $CD^2$ (Jin et al., 2024) addresses knowledge conflict in the decoding phase but does not perform adjustment of conflicts within the representation space itself.

**Representation Editing**  Modifying the representation space of LLMs to adjust their knowledge or response behavior has recently gained significant attention (Liu et al., 2023; Kong et al., 2024; Qiu et al., 2024). Knowledge Neurons (Dai et al., 2022) and ROME (Meng et al., 2022) identify and manipulate specific neurons associated with factual knowledge, injecting or overwriting information directly into the model. While effective for updating long-term knowledge in pretrained models, these methods are less suited for on-the-fly knowledge adjustment during inference. ITI (Li et al., 2023a) detects and modifies representations in real-time by intervening on attention heads, and TruthX (Zhang et al., 2024) explores and adjusts truthfulness signals in both attention and feedforward networks. IRCAN (Shi et al., 2024a) identifies context-sensitive neurons and adjusts their weights to encourage context-faithful generation. Building on this line of work, we leverage a trained autoencoder to explicitly extract knowledge-alignment signals from the representation space and apply them during inference time, mitigating conflicts between internal and retrieved knowledge.

An extended discussion of related work on mechanistic interpretability is provided in Appendix A, and additional analysis comparing our approach with prior work can be found in Appendix B.

## 3 PRELIMINARY

We consider the task of QA, where given an input query $q$, the goal is to select a set of relevant documents from an external corpus $D = \{d_1, d_2, \ldots, d_n\}$ and generate an answer $a = \{a_t\}_{t=1}^m$ based on the retrieved information. In RAG, external knowledge sources are utilized by first retrieving documents relevant to $q$ from $D$ and then generating the response conditioned on the retrieved content (Jin et al., 2025):

$$R = M_D(q), \quad p_{\mathrm{RAG}} = \prod_{t=1}^{m} p_\theta(a_t \mid a_{<t}, q, R), \tag{1}$$

where $M_D$ denotes a retriever that selects relevant documents from $D$ given the query $q$, and $R$ is the resulting set of retrieved documents. $a_{<t}$ represents the partial response sequence preceding the $t$-th token, and $\theta$ denotes the parameters of the language model.

However, when the model's parametric knowledge conflicts with the content of the retrieved documents $R$, it may generate incorrect responses (Tan et al., 2024; Xie et al., 2024). Additionally, if the retrieved set $R$ includes irrelevant information, the injected noise can further degrade output quality (Dai et al., 2024; Su et al., 2025). In standard RAG settings, retrieved documents are treated as additional conditioning input, with limited mechanisms for detecting or resolving conflicts with the model's internal knowledge (Longpre et al., 2021; Tan et al., 2024). This setup implicitly assumes that the model will always trust the retrieved documents, without prioritizing among potentially conflicting sources. Thus, beyond naive retrieval, there is a need for mechanisms that explicitly regulate the knowledge interaction within the model's internal representation space. Such mechanisms should serve to preemptively mitigate conflicts and guide generation toward responses grounded in the most reliable source.

## 4 METHODOLOGY

We aim to mitigate knowledge conflicts in RAG scenarios and promote reliable, document-grounded response generation. We introduce a conflict-aware representation adjustment mechanism, designing an architecture that enables real-time modulation of internal representations. Specifically, we train an autoencoder to reconstruct a separable latent space from intermediate representations extracted from the model layers, capturing the differences between cases where correct and incorrect answers are generated. At inference time, this learned structure adjusts representations, thereby reducing representational distortion under knowledge conflict and guiding the model toward more accurate responses.

### 4.1 LAYER-WISE AUTOENCODER LEARNING

The primary objective of the autoencoder is to map the internal representations of the LLM into a latent space via the encoder and reconstruct them through the decoder (Wang et al., 2016). We leverage this mechanism to identify regions within the model's internal representation space where knowledge conflicts are likely to occur (See Appendix D for AUC; §4.2 for $\delta$).

To train the autoencoder, we construct training samples using existing QA datasets. For each query $q$, we generate two types of samples: positive and negative. Each sample consists of the same system/user instructions, query, and retrieved documents—the only difference lies in the response. The positive answer $a^+$ corresponds to the ground-truth answer. We construct the negative answers from conflictual responses that are generated internally by the model, in order to capture its internal conflict patterns. For the negative answer $a^-$, we consider two cases: parametric and noise. The parametric answer is an incorrect response generated by the model using only its internal knowledge, without access to retrieved evidence. The noise answer is an incorrect response produced even when relevant documents (i.e., documents that provide evidence supporting the ground-truth answer) are retrieved, typically caused by noisy information. This setup allows the autoencoder to learn to distinguish between knowledge-aligned and knowledge-conflicted representations. The details on the construction of the training samples are in Appendix C.

The autoencoder is trained independently of the LLM parameters and operates externally, tailored to the representation characteristics of each layer. At each layer, a token representation $h_t \in \mathbb{R}^d$ (where $t$ indexes the token position in a sequence of length $T$) is compressed into a smaller latent vector $z_t \in \mathbb{R}^{d'}$ through a multi-layer MLP encoder, i.e., $z_t = \mathcal{E}(h_t)$, where $\mathcal{E}(\cdot)$ denotes the encoder network. This process is designed to suppress irrelevant information while preserving essential semantic features (Hinton & Salakhutdinov, 2006; Kingma & Welling, 2013). As a result, representations that vary based on context are mapped to distinct regions in the latent space (Dai et al., 2022). This separation allows the identification of knowledge conflict regions—cases where a single representation reflects conflicting knowledge depending on context, manifested as divergence in latent space. The decoder, also implemented as a multi-layer MLP, reconstructs $\hat{h}_t$ from $z_t$ to match the original dimensionality of $h_t$.

We design the training process around two complementary objectives:

**Reconstruction Loss** The model should reconstruct the input representation. By minimizing the L2 distance between the original input $h$ and the reconstructed output $\hat{h}$, we encourage the model to retain useful features in the compressed latent space. This objective is captured using a mean squared error (MSE)–based reconstruction loss:

$$\mathcal{L}_{\text{recon}} = \frac{1}{T} \sum_{i=1}^{T} \|h_t - \hat{h}_t\|_2^2. \tag{2}$$

**Supervised Contrastive Loss** We aim to separate the representations of correct and incorrect responses, making them distinguishable in the latent space. To achieve this, we incorporate a supervised contrastive loss. For each training sample, we maintain both positive and negative representations and encourage the model to embed positives closer together while pushing negatives further apart. This objective is formulated as:

$$\mathcal{L}_{\text{supcon}} = \sum_i -\log \frac{\sum_{z^+ \in Z_i^+} \exp\big(\text{sim}(z_i, z^+)/\tau\big)}{\sum_{z^+ \in Z_i^+} \exp\big(\text{sim}(z_i, z^+)/\tau\big) + \sum_{z^- \in Z_i^-} \exp\big(\text{sim}(z_i, z^-)/\tau\big)}, \tag{3}$$

where $\text{sim}$ denotes the cosine similarity, $\tau$ is a temperature parameter, $Z_i^+$ is the set of embeddings that share the same class label with $z_i$ (excluding $z_i$ itself), and $Z_i^-$ is the set of embeddings that belong to different classes. The loss encourages an anchor representation $z_i$ to be closer to its positive samples $Z_i^+$ while being pushed apart from negative samples $Z_i^-$.

The losses are combined into the final objective function:

$$\mathcal{L}_{\text{total}} = \mathcal{L}_{\text{recon}} + \mathcal{L}_{\text{supcon}}. \tag{4}$$

### 4.2 INFERENCE-TIME ACTIVATION ADJUSTMENT

The trained autoencoder is incorporated into the model's forward pass during inference, enabling dynamic correction of latent representations as responses are generated. To achieve this, we identify a direction vector $\delta$ in the latent space that mitigates knowledge conflicts. Specifically, $\mathbb{Z}_{\text{pos}}$ and $\mathbb{Z}_{\text{neg}}$ denote the average latent representations of positive and negative samples, respectively, and the direction vector $\delta$ is defined as their difference:

$$\mathbb{Z}_{\text{pos}} = \frac{1}{|D_{\text{pos}}|} \sum_{h \in D_{\text{pos}}} z, \quad \mathbb{Z}_{\text{neg}} = \frac{1}{|D_{\text{neg}}|} \sum_{h \in D_{\text{neg}}} z, \quad \delta = \mathbb{Z}_{\text{pos}} - \mathbb{Z}_{\text{neg}}, \tag{5}$$

where $D_{\text{pos}}$ and $D_{\text{neg}}$ denote the sets of positive and negative training samples, respectively.

We select the top five layers with the most well-separated latent spaces for representation adjustment[1]. For each incoming input, we extract the intermediate representation $h_t$ from each selected layer and pass it through the encoder of the trained autoencoder. To incorporate conflict-mitigation signals, the latent vector is shifted along the learned direction $\delta$, yielding the adjusted latent code $\tilde{z}_t = z_t + \delta$. The corrected latent representation $\tilde{z}_t$ is then passed through the decoder to produce a residual correction term, which is added back to the original representation:

$$\tilde{h}_t = h_t + \lambda \cdot \mathcal{D}(\tilde{z}_t), \tag{6}$$

where $\mathcal{D}(\cdot)$ denotes the decoder, whose output is a one-dimensional vector in the hidden space, and $\lambda$ is a hyperparameter that controls the strength of the correction. In summary, CARE improves the robustness of RAG responses by adjusting the model's internal representations during inference, using conflict-aware signals learned during training.

## 5 EXPERIMENTS

### 5.1 EXPERIMENTAL SETUP

**Datasets** To comprehensively evaluate the effectiveness of CARE, we conduct experiments on six QA datasets. For standard QA, we use Natural Questions (NQ) (Kwiatkowski et al., 2019), TriviaQA (Joshi et al., 2017), PopQA (Mallen et al., 2023), and SQuAD (Rajpurkar et al., 2016). For

---

[1]This is measured by the area under the ROC curve (AUC), and further details are provided in Appendix D. Experimental validation of the number of selected layers is provided in § 5.3.

multi-hop QA, where reasoning over multiple documents is required, we include HotpotQA (Yang et al., 2018) and 2WikiMultiHopQA (2WIKI) (Ho et al., 2020). We use the NQ dataset to train our method, a widely used open-domain QA benchmark consisting of questions extracted from real user search logs and covering a broad range of answer types. During training, we use only the training split of NQ[2]; for evaluation, we use only the test split. We apply a preprocessing step to ensure no overlap between training and test samples. The remaining datasets are used to assess the generalizability of our method. For all datasets, we prioritize using the official test set; if unavailable, we fall back on the evaluation split. We randomly sample 600 examples from each dataset for evaluation (Kim et al., 2024). To measure the accuracy of generated responses, we report Exact Match (EM) and F1 scores[3]. Further details on the dataset are provided in Appendix E.

**Baselines** To evaluate the generalizability of our approach, we consider our method using four instruction-tuned language models: LLaMA-2-7B-Chat (Touvron et al., 2023), LLaMA-2-13B-Chat (Touvron et al., 2023), LLaMA-3.1-8B-Instruct (Llama Team, 2024), and Qwen-2.5-7B-Instruct (Qwen et al., 2025). We evaluate thirteen baselines, categorized into four groups: (1) *Retrieval Usage*: No Retrieval, With Retrieval; (2) *RAG Methods*: Self-RAG (Asai et al., 2023), LLMLingua (Jiang et al., 2023a), Selective-Context (Li et al., 2023b), SURE (Kim et al., 2024), CtrlA (Huanshuo et al., 2025); (3) *Inference-Time Truthfulness Methods*: ITI (Li et al., 2023a), TruthX (Zhang et al., 2024); (4) *Conflict Resolution Methods*: Opin (Jeong et al., 2024), CAD (Shi et al., 2024c; Jin et al., 2024), IRCAN (Shi et al., 2024a), SPARE (Zhao et al., 2025b). However, Self-RAG is based on LLaMA-2-7B, while for ITI and TruthX only versions trained on LLaMA-2-7B-Chat are publicly available, making a rigorous comparison with other models difficult. Accordingly, we use LLaMA-2-7B-Chat to ensure a fair comparison with these methods. Further details on the baselines are provided in Appendix F.

**Implementation Details** All models are implemented using HuggingFace[4], and we utilize the following checkpoints: 'meta-llama/Llama-2-7b-chat-hf', 'meta-llama/Llama-2-13b-chat-hf', 'meta-llama/Llama-3.1-8B-Instruct', and 'Qwen/Qwen2.5-7B-Instruct'. Training is performed using 8 NVIDIA RTX A6000 48GB GPUs with the AdamW optimizer, $1 \times 10^{-4}$ learning rate, and 10 epochs. Inference is conducted on a single GPU with a maximum input length of 2048 tokens and a maximum output length of 32 tokens[5]. For retrieval, we adopt E5-base-v2 (Wang et al., 2024) as the default retriever, using the 'intfloat/e5-base-v2' checkpoint. We retrieve five passages per query from the DPR version of the Wikipedia dataset (December 2018 snapshot). For additional experiments, BM25 (Robertson et al., 2009) is implemented using Pyserini (Lin et al., 2021). The latent dimension $z$ of the autoencoder is set to one-quarter of the hidden size, following Zhang et al. (2024), and is implemented as a 2-layer MLP. The editing strength $\lambda$ is tuned for each model: 0.3 for LLaMA-2-7B-Chat and LLaMA-3.1-8B-Instruct, and 0.5 for both LLaMA-2-13B-Chat and Qwen-2.5-7B-Instruct. Most baseline experiments are conducted using the FlashRAG framework (Jin et al., 2025). We train and evaluate IRCAN using the official implementation on github[6]. Because of out-of-memory issues, training is conducted on 8 NVIDIA SXM4 A100 80GB GPUs. The model is trained on the same dataset as used in our method for fair comparison. Further implementation details are provided in Appendix G, and the prompt templates used are provided in Appendix H.

## 5.2 EXPERIMENTAL RESULTS

Table 1 presents a quantitative evaluation of CARE across various instruction-tuned LLM backbones and QA benchmarks. In particular, Table 2 reports results with LLaMA-2-7B-Chat, comparing CARE against Self-RAG, ITI, and TruthX. Overall, when applied to different LLMs, CARE consistently achieves the highest average performance in RAG settings across diverse QA datasets. These results lead to the following findings:

---

[2] We experimentally demonstrate that training solely on NQ suffices to ensure generalization; results from training on HotpotQA are reported in § 5.3.

[3] EM and F1 are valid for assessing response correctness and evidence integration in RAG, while finer effects of knowledge conflicts are examined in § 5.3.

[4] https://huggingface.co/

[5] LLaMA-2-13B-Chat tends to generate overly long responses; therefore, we set the maximum output length to 64 tokens.

[6] https://github.com/danshi777/IRCAN

Table 1: Performance across QA datasets for various methods and LLM backbones. Best results per model group are in **bold**.

| Method | NQ EM | NQ F1 | TriviaQA EM | TriviaQA F1 | PopQA EM | PopQA F1 | SQuAD EM | SQuAD F1 | HotpotQA EM | HotpotQA F1 | 2WIKI EM | 2WIKI F1 | AVG |
|---|---|---|---|---|---|---|---|---|---|---|---|---|---|
| **LLaMA-2-13B-Chat** | | | | | | | | | | | | | |
| No Retrieval | 13.33 | 19.27 | 33.67 | 39.88 | 9.67 | 19.52 | 6.67 | 11.11 | 11.33 | 18.42 | 14.50 | 18.79 | 18.01 |
| With Retrieval | 29.67 | 44.00 | 42.00 | 57.84 | 25.33 | 38.89 | 19.17 | 31.90 | 16.67 | 28.97 | 16.00 | 26.04 | 31.37 |
| LLMLingua | 26.17 | 39.19 | 40.67 | 54.75 | 24.83 | 37.96 | 15.50 | 28.66 | 16.83 | 28.85 | 16.00 | 24.70 | 29.51 |
| Selective-Context | 21.17 | 34.01 | 33.67 | 48.30 | 13.33 | 29.07 | 11.00 | 23.46 | 13.67 | 24.61 | 14.00 | 23.74 | 24.17 |
| SURE | 29.17 | 42.21 | **50.67** | 59.48 | 35.50 | 42.72 | 19.33 | 29.75 | **19.17** | 29.01 | 18.67 | 27.06 | 33.56 |
| Opin | 15.33 | 24.14 | 27.50 | 43.38 | 12.83 | 22.62 | 7.17 | 15.13 | 7.50 | 16.18 | 3.33 | 16.76 | 17.66 |
| CAD | 29.00 | 43.17 | 41.00 | 57.25 | 23.50 | 37.34 | 17.50 | 30.85 | 16.17 | 28.63 | 14.83 | 25.15 | 30.37 |
| IRCAN | 33.67 | 46.36 | 47.50 | 59.16 | **37.83** | **44.37** | 18.17 | 31.65 | 19.00 | 29.19 | **19.50** | 28.58 | 34.58 |
| CARE | **34.00** | **47.09** | 49.00 | **62.23** | 33.50 | 43.37 | **19.83** | **33.17** | 18.33 | **30.26** | 17.17 | **28.71** | **34.72** |
| **LLaMA-3.1-8B-Instruct** | | | | | | | | | | | | | |
| No Retrieval | 26.67 | 37.53 | 52.17 | 60.15 | 24.17 | 29.17 | 12.00 | 21.39 | 19.00 | 28.06 | 15.83 | 27.18 | 29.44 |
| With Retrieval | 35.83 | 49.64 | 58.83 | 68.72 | 39.67 | 48.67 | 29.50 | 40.58 | 27.50 | 38.41 | 14.33 | 25.06 | 39.73 |
| LLMLingua | 34.33 | 47.38 | 58.17 | 67.90 | 37.00 | 44.89 | 27.00 | 38.40 | 27.33 | 38.29 | **22.83** | **30.82** | 39.53 |
| Selective-Context | 32.67 | 45.71 | 57.83 | 67.14 | 36.00 | 42.89 | 23.00 | 32.96 | 25.50 | 35.19 | 17.67 | 26.32 | 36.91 |
| SURE | 36.17 | 51.14 | 48.00 | 62.28 | **41.50** | 47.98 | 24.33 | 37.15 | 19.17 | 32.52 | 9.00 | 15.49 | 35.39 |
| Opin | 38.17 | 47.78 | 58.83 | 68.34 | 38.00 | 43.69 | 30.00 | 39.72 | 25.00 | 33.77 | 12.00 | 21.18 | 38.04 |
| CAD | 36.67 | 49.60 | 57.67 | 68.47 | 38.83 | 48.22 | 28.50 | 40.41 | 28.33 | 39.13 | 16.83 | 26.99 | 39.97 |
| IRCAN | 37.83 | 49.55 | 50.67 | 61.56 | 38.17 | 45.82 | 21.17 | 35.67 | 23.17 | 33.65 | 9.17 | 19.71 | 35.51 |
| CARE | **38.50** | **51.31** | **59.67** | **69.96** | 40.33 | **48.84** | **30.33** | **41.85** | **29.50** | **40.36** | 18.33 | 27.51 | **41.37** |
| **Qwen-2.5-7B-Instruct** | | | | | | | | | | | | | |
| No Retrieval | 15.83 | 24.14 | 43.50 | 49.74 | 17.17 | 21.56 | 11.33 | 20.19 | 20.33 | 28.40 | 24.83 | 30.65 | 25.64 |
| With Retrieval | 37.00 | 49.12 | 61.67 | 69.91 | 40.17 | 47.59 | 31.17 | 41.61 | 29.17 | 40.13 | 25.00 | 32.06 | 42.05 |
| LLMLingua | 30.00 | 41.78 | 57.33 | 66.95 | 35.17 | 42.91 | 24.50 | 35.60 | 27.67 | 38.00 | 26.67 | 32.15 | 38.23 |
| Selective-Context | 31.50 | 44.26 | 56.33 | 64.47 | 34.00 | 40.09 | 21.67 | 32.23 | 25.33 | 35.13 | 24.67 | 31.66 | 36.78 |
| SURE | **45.17** | **55.17** | 62.00 | 69.60 | **42.67** | 47.67 | 30.67 | 38.78 | 27.00 | 36.14 | 17.67 | 23.60 | 41.35 |
| Opin | 37.50 | 49.00 | 61.00 | 69.22 | 41.50 | 47.97 | 33.00 | 42.17 | 30.17 | 40.30 | 25.67 | 31.99 | 42.46 |
| CAD | 37.00 | 49.00 | 61.33 | 69.61 | 40.33 | 47.81 | 31.00 | 40.87 | 29.67 | 40.51 | 25.33 | 32.63 | 42.09 |
| IRCAN | 36.33 | 47.91 | 60.17 | 68.63 | 40.33 | 47.39 | 29.67 | 40.33 | 29.17 | 39.26 | 24.00 | 31.60 | 41.23 |
| CARE | 40.17 | 51.37 | **62.17** | **70.37** | 41.67 | **48.43** | **34.17** | **44.29** | **32.00** | **42.84** | **28.00** | **35.73** | **44.27** |

Table 2: Performance across QA datasets for LLaMA-2-7B-Chat. Self-RAG and SPARE both use LLaMA-2-7B as their backbone. PK denotes *steer to use parametric knowledge*, and CK denotes *steer to use contextual knowledge*.

| Method | NQ EM | NQ F1 | TriviaQA EM | TriviaQA F1 | PopQA EM | PopQA F1 | SQuAD EM | SQuAD F1 | HotpotQA EM | HotpotQA F1 | 2WIKI EM | 2WIKI F1 | AVG |
|---|---|---|---|---|---|---|---|---|---|---|---|---|---|
| No Retrieval | 12.50 | 22.04 | 37.33 | 46.40 | 6.33 | 15.09 | 6.33 | 15.09 | 15.33 | 24.08 | **25.17** | **31.67** | 21.45 |
| With Retrieval | 16.33 | 29.24 | 42.33 | 55.65 | 28.50 | 38.05 | 8.33 | 20.18 | 15.33 | 25.87 | 14.83 | 26.39 | 26.75 |
| Self-RAG | **40.33** | **48.54** | 41.50 | 56.56 | 22.17 | 34.38 | 16.00 | 28.73 | 15.00 | 28.32 | 12.67 | 26.03 | 30.85 |
| CtrlA | 13.72 | 18.28 | 25.25 | 33.94 | 12.76 | 13.16 | **43.73** | **47.09** | **19.66** | 27.10 | 21.40 | 27.21 | 25.78 |
| SPARE_PK | 26.67 | 35.41 | 50.33 | 58.70 | 34.83 | 39.48 | 14.33 | 23.33 | 7.00 | 10.13 | 15.67 | 18.49 | 27.86 |
| SPARE_CK | 4.50 | 7.53 | 9.00 | 12.75 | 1.83 | 3.97 | 0.83 | 2.84 | 17.50 | 24.29 | 20.50 | 27.73 | 11.11 |
| ITI | 0.33 | 7.87 | 2.83 | 14.49 | 2.33 | 11.85 | 0.17 | 6.69 | 0.50 | 5.82 | 0.00 | 7.25 | 5.01 |
| TruthX | 15.00 | 29.39 | 43.83 | 58.04 | 32.83 | 41.55 | 10.17 | 23.12 | 15.33 | 27.10 | 13.00 | 27.28 | 28.05 |
| CARE | 25.33 | 37.94 | **53.17** | **62.49** | 40.67 | 46.51 | 14.17 | 26.63 | 19.50 | **28.95** | 21.50 | 30.28 | **33.93** |

**Existing RAG optimization methods fail to provide generalized performance across diverse benchmarks.** Methods such as LLMLingua and Selective-Context primarily focus on input compression or context selection. While these strategies effectively reduce redundancy, they are insufficient for handling conflicting evidence. SURE achieves competitive results on specific single-hop QA tasks such as PopQA; however, on multi-hop benchmarks (HotpotQA, 2WIKI), its performance falls behind simple retrieval, revealing limitations in more complex reasoning scenarios. Self-RAG and CtrlA improve performance on NQ and PopQA, respectively, but their average scores remain below that of CARE (33.93), indicating that their benefits are not consistent across datasets.

**Existing knowledge conflict resolution methods are fragile in RAG settings.** Methods such as CAD and IRCAN, which are designed for conflict resolution without retrieval, do not explicitly account for noisy evidence. As a result, their performance deteriorates significantly when applied to RAG. For example, on LLaMA-3.1-8B-Instruct, CAD and IRCAN achieve average scores of

Table 3: Ablation results across four QA datasets. The values in parentheses indicate performance changes relative to the full method. $\mathcal{L}_{recon}$ denotes the reconstruction loss, and $\mathcal{L}_{supcon}$ represents the supervised contrastive loss. $\delta$ is the conflict mitigation direction vector.

| Method | NQ | | TriviaQA | | PopQA | | SQuAD | |
|---|---|---|---|---|---|---|---|---|
| | EM | F1 | EM | F1 | EM | F1 | EM | F1 |
| CARE | 38.50 | 51.31 | 59.67 | 69.96 | 40.33 | 48.84 | 30.33 | 41.85 |
| w/o $\mathcal{L}_{recon}$ | 0.50 (↓38.00) | 3.08 (↓48.23) | 1.00 (↓58.67) | 5.63 (↓64.33) | 0.83 (↓39.50) | 3.46 (↓45.38) | 0.67 (↓29.67) | 3.21 (↓38.64) |
| w/o $\mathcal{L}_{supcon}$ | 37.17 (↓1.33) | 49.33 (↓1.98) | 59.17 (↓0.50) | 69.12 (↓0.84) | 39.67 (↓0.67) | 48.45 (↓0.40) | 29.50 (↓0.83) | 40.75 (↓1.10) |
| $-\delta$ | 28.33 (↓10.17) | 43.51 (↓7.80) | 48.50 (↓11.17) | 62.08 (↓7.88) | 28.83 (↓11.50) | 41.24 (↓7.62) | 24.50 (↓5.83) | 36.68 (↓5.17) |
| Random $\delta$ | 22.17 (↓16.33) | 36.86 (↓14.45) | 36.33 (↓23.33) | 50.81 (↓19.15) | 19.50 (↓20.83) | 31.56 (↓17.28) | 17.33 (↓13.00) | 29.80 (↓12.05) |

39.97 and 35.51, respectively, which are comparable to or worse than simple retrieval (39.73). Opin attains strong performance on Qwen-2.5-7B-Instruct with an average of 42.46, but shows substantially lower scores on LLaMA-based models, indicating limited cross-architecture generalization. SPARE tends to improve performance when parametric knowledge is amplified, but this also leads to results that are biased toward a particular knowledge source. Truthfulness-oriented methods such as TruthX and ITI perform poorly on LLaMA-2-7B-Chat, with averages of 28.05 and 5.01, respectively, demonstrating that they are primarily ineffective in noisy, conflict-prone RAG environments.

**CARE achieves consistent improvements under noisy retrieval environments.** Our proposed method achieves the highest average performance across all model families. On LLaMA-3.1-8B-Instruct, CARE reaches an average of 41.37, surpassing the strongest baseline CAD (39.97). On LLaMA-2-13B-Chat, it records 34.72, slightly higher than the best baseline IRCAN (34.58). Furthermore, on Qwen-2.5-7B-Instruct, CARE achieves 44.27, confirming that the observed gains generalize across different architectures. Moreover, although CARE is trained only on the NQ dataset, it still surpasses the baselines on other datasets, indicating that it learns a conflict-mitigation direction that generalizes across tasks. Notably, the improvements are pronounced not only on single-hop QA but also on multi-hop QA benchmarks, demonstrating that CARE maintains knowledge consistency even in complex reasoning scenarios beyond simple pattern matching. These results establish CARE as a robust and generalizable approach that enhances reliability in RAG, outperforming existing RAG and conflict resolution techniques under noisy and conflicting retrieval conditions.

We discuss the random seeds used for evaluation data sampling, the performance on medical and legal domains, and the results on Qwen3-4B-Thinking (Yang et al., 2025) model in Appendix I.

## 5.3 ANALYSIS

We conduct a series of analysis experiments to investigate the impact of various components in our method. We use NQ and LLaMA-3.1-8B-Instruct for analyses, considering space constraints.

**Ablation Study** Table 3 presents the ablation studies designed to assess the contribution of each component within CARE. We first examine the impact of removing individual training losses. Excluding the reconstruction loss $\mathcal{L}_{recon}$ results in a substantial drop in both EM and F1 scores across all datasets. This indicates that preserving the input-output consistency in the autoencoder is critical for learning stable internal representations. Removing the supervised contrastive loss $\mathcal{L}_{supcon}$ also leads to performance degradation, confirming that contrastive learning helps the model distinguish between knowledge-aligned and conflicting representations, thereby enhancing representational robustness. To further evaluate the role of the direction vector $\delta$, we perform two additional experiments: one where the direction is reversed ($-\delta$), and another where a random direction is applied (Random $\delta$). Performance drops in the $-\delta$, suggesting that the method's effectiveness depends on using the correct directional adjustment in latent space. The Random $\delta$ results in even larger performance drops, emphasizing the importance of learning a meaningful and targeted adjustment vector for effective knowledge conflict mitigation.

**Resolution of Conflicts Induced by External Knowledge** To more rigorously assess whether knowledge conflicts are resolved, we conduct a multifaceted analysis of how external knowledge affects model responses, as shown in Table 4. We report the average EM across NQ, TriviaQA, PopQA, and SQuAD. Retention denotes cases where the model's

Table 4: Analysis of knowledge conflict resolution with external knowledge.

| Method | Retention ↑ | Resolution ↑ | Conflict ↓ |
|---|---|---|---|
| With Retrieval | **5.83** | 35.13 | 6.17 |
| SURE | 4.92 | 32.58 | 7.08 |
| IRCAN | 5.17 | 31.79 | 6.83 |
| CARE | 5.79 | **36.42** | **6.13** |

Table 5: Performance comparison of `CARE` when trained on NQ and HotpotQA.

| Trained Dataset | NQ | | TriviaQA | | PopQA | | SQuAD | | HotpotQA | | 2WIKI | | AVG |
|---|---|---|---|---|---|---|---|---|---|---|---|---|---|
| | EM | F1 | EM | F1 | EM | F1 | EM | F1 | EM | F1 | EM | F1 | |
| NQ | 38.50 | 51.31 | 59.67 | 69.96 | 40.33 | 48.84 | 30.33 | 41.85 | 29.50 | 40.36 | **18.33** | **27.51** | 41.37 |
| HotpotQA | **42.00** | **53.28** | **63.67** | **72.06** | **44.33** | **50.54** | **34.50** | **44.82** | **29.83** | **39.93** | 17.50 | 25.84 | **43.19** |

parametric knowledge remains correct and stable even after incorporating noisy external documents. Resolution refers to cases where the model initially fails to answer correctly but succeeds once external knowledge is added, indicating successful conflict resolution. Conflict captures cases where a correct parametric answer is overturned into an incorrect one after external knowledge is introduced, reflecting unresolved or newly induced conflicts. Retention score is almost identical to that of With Retrieval, indicating that, unlike other methods, it very rarely degrades performance on instances where the model already produces the correct answer. At the same time, CARE achieves the highest Resolution, demonstrating its ability to supplement incomplete parametric knowledge with retrieved information effectively. The lowest Conflict indicates that CARE does not corrupt answers that were already correct based solely on parametric knowledge, even in the presence of retrieval-induced noise. These results suggest that CARE internally selects the appropriate source of truth and improves accuracy through effective knowledge integration while minimizing side effects.

**Impact of Hyperparameter**  Figure 2 illustrates the impact of different hyperparameter settings on model performance. In the left plot, we observe that increasing the value of $\lambda$ leads to a sharp decline in scores. This suggests that while moderate perturbation is beneficial for adjusting representations, excessive perturbation adversely impacts representation editing. Based on this observation, we set $\lambda = 0.3$, which yields the highest F1 score. The right plot shows the effect of varying the number of model layers used for representation editing. Using the top 5 most informative layers achieves the best performance, indicating that minimal yet targeted modifi-

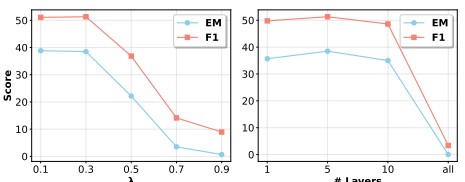

Figure 2: Performance variation with respect to $\lambda$ (left) and the number of edited layers (right). $\lambda$ denotes the strength of representation editing, while # Layers indicates the number of layers in which editing is applied.

cations to key layers are sufficient to mitigate knowledge conflicts. In contrast, applying edits to all layers leads to a notable performance drop, underscoring that over-intervention can degrade output quality and highlighting the importance of careful hyperparameter selection.

**Effect of Retrieval Settings**  Figure 3 compares performance under different retrieval strategies (E5 and BM25), and varying numbers of retrieved documents. E5, the primary retriever used in our work, consistently outperforms BM25 in retrieval quality. However, even under the noisier BM25, `CARE` maintains comparable performance gains, demonstrating its robustness and potential for generalization across retrieval conditions. While performance is similar when retrieving only $k = 1$ document, increasing $k$ consistently improves performance across all settings. This suggests that incorporating more documents, even with some noise, can lead to more robust responses, as `CARE` effectively mitigates conflicts.

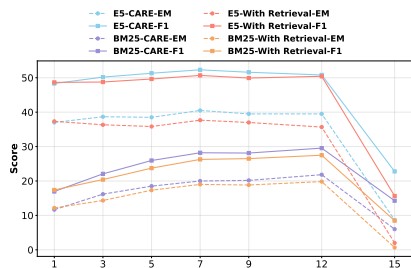

Figure 3: Performance comparison under different retrieval methods and varying numbers of retrieved documents $k$.

**Effect of Training on Alternative Dataset**  To evaluate the generalizability and transferability of `CARE`, we compare its performance when trained on two different datasets: NQ, which is used as the primary training dataset in our main experiments, and HotpotQA, a more challenging multi-hop QA benchmark. We report results across the same six evaluation benchmarks, as summarized in Table 5. Training `CARE` on HotpotQA leads to improved EM and F1 scores across nearly all evaluation datasets compared to training on NQ. Notably, EM scores improve by 3.5 on NQ, 4.0 on TriviaQA, and 4.0 on PopQA. We also observe consistent gains on SQuAD. These improvements suggest that training on a more challenging multi-hop dataset like HotpotQA helps the model generalize better to relatively more straightforward QA tasks. The performance on HotpotQA itself is preserved, and although there is a slight drop on 2WIKI, the overall average score increases from 41.37 to

43.19. These results indicate that CARE does not overfit to a specific dataset but instead generalizes effectively across diverse QA domains.

**Visualization of Knowledge Conflict Space**   Figure 4 shows the 2D projection of positive and negative sample representations using PCA. The visualization reveals that the two groups form separable clusters in the representation space, indicating that our training method effectively disentangles representations based on the presence or absence of knowledge conflict. The positive samples exhibit a more compact structure, suggesting that the model has learned consistent representations grounded in aligned information.

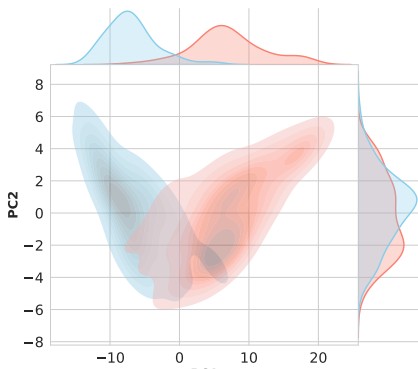

Figure 4: 2D projection of representations for positive (blue) and negative (red) samples.

**Efficiency Analysis**   Table 13 reports average runtime, decomposed into retrieval and inference latency. CARE achieves a total inference time of 1.83s, nearly identical to the baseline With Retrieval (1.87s), showing that the co-ordination mechanism introduces negligible latency. This efficiency arises because CARE does not spend additional time generating evasive or uncertain responses, but instead produces clear and direct answers to the query. By contrast, compression-based methods (LLMLingua, Selective-Context) introduce pre-processing latency, whereas CAD and IR-CAN incur higher inference costs due to the complexity of post-retrieval reasoning. SURE shows the largest slowdown, with an average total time exceeding 7s. Overall, CARE achieves superior accuracy while maintaining efficiency comparable to standard RAG, striking a favorable balance between reliability and deployment cost.

We present additional ablations of our method, layer-wise AUC analyses, and the distribution of negative samples in Appendix J, and provide case studies in Appendix K.

# 6   CONCLUSION

In this work, we proposed CARE, a novel method designed to address conflicts between the parametric knowledge and externally retrieved information. By analyzing neuron activation patterns, CARE identifies points of knowledge conflict and employs an autoencoder-based architecture to suppress irrelevant or contradictory signals. Experimental results across various models and benchmarks demonstrate that CARE outperforms existing RAG and knowledge mitigation methods regarding answer accuracy. These findings highlight the potential of representation-level conflict resolution as a promising direction for future research in reliable, knowledge-grounded response generation.

## LIMITATIONS

While CARE consistently improves robustness in retrieval-augmented QA by mitigating conflicts between parametric and retrieved knowledge, it still has the following limitations.

- CARE is primarily trained on datasets where retrieval-induced confusion within long contexts is the dominant source of errors. As a result, gains can be more modest on benchmarks such as PopQA, where failures are driven more by straightforward factual verification or parametric inaccuracies than by knowledge conflicts. Future work should explore training regimes that adapt the mixture and type of negative samples to the characteristics of the target domain.

- CARE applies a single, pre-learned direction vector uniformly to all inputs at inference time. While this global intervention is generally safe and yields substantial gains compared to not applying it, it can introduce slight distortions when the model's parametric knowledge is already correct. A promising extension is to make the intervention strength adaptive to the estimated conflict level.

- The choice of editing layers and intervention strength currently relies on empirical procedures such as AUC-based layer scoring and grid search. Developing more principled, automated methods to select effective intervention points and magnitudes remains an important direction for future work.

ETHICAL STATEMENT

This work raises no specific ethical issues. It excludes human and animal subjects as well as any personal or sensitive data. The datasets utilized are publicly available and legally licensed for academic research. The modeling approaches and experimental methods strictly adhere to community-recognized ethical standards and are not designed for harmful, malicious, or discriminatory outcomes. Accordingly, we believe this research upholds the principles of responsible AI and entails no foreseeable risks of misuse.

REPRODUCIBILITY STATEMENT

All experiments were conducted using the open-source framework provided by FlashRAG (Jin et al., 2025). Experiments were primarily executed on a workstation equipped with 8 NVIDIA RTX A6000 48GB GPUs. Details of the experimental setup and implementation—including model parameter counts, total computational budget, and the best-found hyperparameter values—are provided in § 5.1 and Appendix G. Details of the hyperparameter search are provided in § 5.3 and Appendix J, and all prompt templates are listed in Appendix H.

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

## A    MORE RELATED WORK

**Mechanistic Interpretability**    Recent mechanistic interpretability work has leveraged sparse dictionary learning (SDL) and Sparse Autoencoders (SAEs) to decompose internal LLM representations into interpretable, monosemantic features (Bricken et al., 2023; Cunningham et al., 2023; Gao et al., 2024). These approaches typically rely on overcomplete latent spaces and strong sparsity regularization to disentangle complex activation patterns. Building on this line of work, SPARE (Zhao et al., 2025b) further demonstrated that SAE-derived latent features can be directly manipulated to steer knowledge-selection behaviors in LLMs, highlighting the controllability and interpretability enabled by sparse latent structures. Although our method also employs an autoencoder architecture to analyze representation space, its purpose differs fundamentally from that of SAEs. Instead of pursuing unsupervised feature decomposition for interpretability, we utilize a compressed, dense latent space trained with supervised contrastive signals. This enables us to identify and manipulate global directions associated with knowledge conflict, prioritizing the robustness and controllability of model behavior.

## B    COMPARISON WITH RELATED WORK

While our work is related to various RAG and representation-editing approaches in that it aims to mitigate knowledge conflicts caused by imperfect retrieval, it differs markedly in both goals and design.

- ASTUTE RAG (Wang et al., 2025): Building on the observation that incorrect or even adversarial content is inevitably introduced by imperfect retrieval, ASTUTE RAG repeatedly integrates internal knowledge and external documents in a source-aware manner and selects the final answer based on reliability. In contrast, CARE is not tied to any particular RAG algorithm; instead, it directly intervenes in the intermediate representations of the LLM after retrieval has already been performed, weakening conflict signals between parametric knowledge and document-based knowledge via representation editing. Thus, ASTUTE RAG operates at the high-level design of RAG strategies, whereas CARE performs fine-grained adjustments in the representation space of a given LLM, making the two approaches complementary.

- CtrlA (Huanshuo et al., 2025): Although CtrlA also approaches RAG from a representation perspective, its goal is *Adaptive RAG*, namely deciding when to trigger retrieval. CtrlA extracts honesty and confidence directions from the LLM's internal representations and uses them to control whether and when retrieval should be performed, proposing an inherent control framework. In contrast, CARE applies a *Robust RAG* perspective: it assumes that retrieval has already been performed and may have injected noise, and then corrects the model's internal state after the fact.

- SPARE (Zhao et al., 2025b): SPARE leverages SAEs to extract sparse features in the LLM's latent space that correspond to parametric and contextual knowledge, and then modulates these features to control which knowledge source is emphasized. SPARE and CARE both manipulate latent representations, but differ in two key aspects. (i) SPARE focuses on adjusting knowledge selection at the level of interpretable sparse features, whereas CARE learns conflict patterns directly in a compressed latent space and moves representations along directions that alleviate those conflicts. (ii) SPARE requires relatively high computational cost for training SAEs and extracting interpretable features, while CARE employs a lightweight autoencoder architecture centered on a 1D direction vector, making it easier to deploy across different models and domains.

## C    TRAINING DATA DESIGN

The training samples for learning the autoencoder are designed to separate and identify latent representations that yield the correct answer to a question—given retrieved documents as evidence—from those that produce confounded responses. As shown in Figure 1, we approach this problem with contrastive learning and, for each question, construct paired samples consisting of a positive answer and a negative answer. Here, the positive answer is the correct answer to the question, while the negative answer is an incorrect one; the two stand in a 1:1 correspondence per question. In particular, nega-

tive answers are extracted from actual model outputs that are wrong and comprise the following two types in equal proportion[7]:

- **Parametric Answer**: An incorrect answer generated solely from the model's internal (parametric) knowledge without retrieval. We obtain these by collecting wrong responses produced when only the question is given to the model.

- **Noise Answer**: An incorrect answer generated despite access to relevant retrieved documents, typically due to noise or contextual confusion. That is, we collect cases where the retrieved documents contain the correct evidence but the model misuses or overlooks it and answers incorrectly.

We include only negative samples in which the model's response is clearly incorrect, as determined by an F1 score of 0.

## D  ROC–AUC ESTIMATION

**Centroid-based Scoring**  Given an arbitrary latent vector $z$, we define a centroid-based score:

$$f(z) = \frac{\|z - \mathbb{Z}_{\text{neg}}\|_2}{\|z - \mathbb{Z}_{\text{pos}}\|_2 + \|z - \mathbb{Z}_{\text{neg}}\|_2}. \tag{7}$$

This score takes values in $[0, 1]$, where larger values indicate that $z$ is closer to the positive centroid.

**ROC and AUC**  Let $\{f(z_i^+)\}_{i=1}^m$ and $\{f(z_j^-)\}_{j=1}^n$ denote the scores for positive and negative samples, respectively. For a threshold $\gamma \in \mathbb{R}$, define

$$\text{TPR}(\gamma) = \frac{1}{m} \sum_{i=1}^m \mathbf{1}[f(z_i^+) \geq \gamma], \qquad \text{FPR}(\gamma) = \frac{1}{n} \sum_{j=1}^n \mathbf{1}[f(z_j^-) \geq \gamma]. \tag{8}$$

The ROC curve is the set $\{(\text{FPR}(\gamma), \text{TPR}(\gamma)) : \gamma \in \mathbb{R}\}$, and AUC is

$$\text{AUC} = \int_0^1 \text{TPR} \, d(\text{FPR}). \tag{9}$$

**Empirical Estimation**  In practice, AUC is computed via the Mann–Whitney statistic:

$$\widehat{\text{AUC}} = \frac{1}{mn} \sum_{i=1}^m \sum_{j=1}^n \Big( \mathbf{1}[f(z_i^+) > f(z_j^-)] + \tfrac{1}{2}\mathbf{1}[f(z_i^+) = f(z_j^-)] \Big). \tag{10}$$

## E  MORE DATASETS

We use the same set of datasets as those employed in FlashRAG, all of which are publicly available via Hugging Face[8]. Detailed descriptions of the datasets used in our experiments are provided below:

- **Natural Questions (NQ)** (Kwiatkowski et al., 2019): A large-scale QA dataset consisting of real user queries collected from Google Search and associated Wikipedia documents. The questions are naturally phrased and diverse, and the answers are extracted from specific paragraphs or sentences within the documents. Because of its emphasis on long-form and context-rich questions, NQ serves as a strong benchmark for evaluating a model's reading comprehension ability.

- **TriviaQA** (Joshi et al., 2017): A QA dataset featuring quiz-style questions paired with evidence documents from the web. The questions are not synthetic but are collected from actual trivia quizzes, requiring complex reasoning and substantial background knowledge to answer correctly.

- **PopQA** (Mallen et al., 2023): A large-scale open-domain QA dataset designed to assess factual knowledge based on Wikidata. It primarily covers commonsense and factual questions across diverse domains such as entertainment, history, and sports.

---

[7] We further validate the composition of negative answers in Appendix J.4.

[8] https://huggingface.co/datasets/RUC-NLPIR/FlashRAG_datasets

- **SQuAD** (Rajpurkar et al., 2016): A widely used QA benchmark focused on paragraph-level reading comprehension over Wikipedia articles. Answers are extracted as continuous text spans from a single paragraph. Its high-quality human-annotated questions and answers make it a standard for evaluating the base performance of QA models.
- **HotpotQA** (Yang et al., 2018): A multi-hop QA dataset that requires reasoning over multiple supporting documents. In addition to the answer, supporting facts are provided to facilitate the evaluation of multi-step reasoning and explainability in model predictions.
- **2WikiMultiHopQA (2WIKI)** (Ho et al., 2020): A multi-hop QA dataset that involves reasoning across multiple Wikipedia articles. Answering a question typically requires integrating information from at least two distinct documents, making it suitable for assessing complex relational and world knowledge reasoning.

We randomly sample 5,000 instances from the original training dataset for evaluation. The evaluation aims to identify negative samples by selecting instances with an F1 score of 0. When the NQ training dataset is used, evaluation with the LLaMA-3.1-8B-Instruct model yields 766 instances.

## F    MORE BASELINES

We provide a detailed description of each adopted baseline method below:

- **No Retrieval**: A baseline setting where the model generates responses solely based on its parametric knowledge, without using any external documents.
- **With Retrieval**: A standard RAG approach where the retrieved documents $R$ are directly appended to the prompt.
- **Self-RAG** (Asai et al., 2023): A self-reflective RAG in which the model itself performs query expansion/rewriting and decides whether external evidence is needed, then—on the retrieved documents—passes through critic, grounding, and refinement stages to produce the final answer. The implementation uses 'selfrag/selfrag_llama2_7b.'
- **LLMLingua** (Jiang et al., 2023a): A prompt compression method that minimizes semantic loss during compression, thereby reducing the possibility of knowledge conflicts caused by irrelevant information and promoting consistent response generation. Following the paper, we use 'meta-llama/Llama-2-7b-hf' as the model for refinement.
- **Selective-Context** (Li et al., 2023b): A context filtering method that removes parts of the LLM input with low self-information, reducing context length and improving processing efficiency without degrading performance. We use 'openai-community/gpt2' as the model for refinement.
- **SURE** (Kim et al., 2024): Summarizes retrieved documents and compares them with candidate answers to select the response that best aligns with the given context, thereby resolving conflicting knowledge.
- **ITI** (Li et al., 2023a): A truthfulness-oriented, chat-model–based approach that detects signals of factual conflict, injects factuality-strengthening guidance, and encourages restraint and self-correction when uncertain. It is designed to prioritize contextual consistency in both the generation and evaluation stages; in our setup, we use 'likenneth/honest_llama2_chat_7B.'
- **CtrlA** (Huanshuo et al., 2025): An adaptive RAG framework that leverages honesty and confidence directions in a model's internal representations to decide whether and when to trigger retrieval.
- **TruthX** (Zhang et al., 2024): A fine-tuned model designed to reduce hallucinations and promote honest responses. It encourages consistency with the given context, verifiability, and explicit uncertainty marking to suppress ungrounded knowledge recall. Our implementation uses 'ICTNLP/Llama-2-7b-chat-TruthX.'
- **Opin** (Jeong et al., 2024): A prompting strategy that reformulates a question into one asking for the speaker's opinion, encouraging the model to prioritize contextual interpretation.
- **CAD** (Shi et al., 2024c; Jin et al., 2024): A contrastive decoding approach that adjusts the difference in token probabilities between generation with and without the given context, guiding the model to produce content grounded in the context rather than relying on prior knowledge, thus mitigating knowledge conflicts.

Table 6: Prompt template used for No Retrieval. The model is instructed to answer the question solely based on its internal knowledge without access to any retrieved context.

---

[SYSTEM PROMPT]
Answer the question based on your own knowledge. Only give
me the answer and do not output any other words.

[USER PROMPT]
Question: {question}
Answer:

---

Table 7: Prompt template used for With Retrieval, LLMLingua, Selective-Context, CAD, IRCAN, and CARE. The model is instructed to answer the question based on the retrieved passages, which are explicitly included in the prompt.

---

[SYSTEM PROMPT]
Answer the question based on the given passage. Only give me
the answer and do not output any other words. The following
are given passages: {retrieval passages}

[RETRIEVAL PASSAGES PROMPT]
Doc 1 (Title: {title}) {content}
Doc 2 (Title: {title}) {content}
Doc 3 (Title: {title}) {content}
Doc 4 (Title: {title}) {content}
Doc 5 (Title: {title}) {content}

[USER PROMPT]
Question: {question}
Answer:

---

- **IRCAN** (Shi et al., 2024a): Identifies neurons that are highly sensitive to contextual information—especially when it conflicts with pre-trained knowledge—via the integrated gradients technique, and reweights the selected neurons to amplify their influence during generation. The paper emphasizes that combining CAD mitigates such conflicts; accordingly, in Appendix J.5, we apply CAD to IRCAN and CARE to verify its effect.
- **SPARE** (Zhao et al., 2025b): Steers LLM behavior by manipulating sparse SAE features that disentangle parametric and contextual knowledge, and in our setup the SAE is instantiated with the publicly released pretrained checkpoint 'yuzhaouoe/Llama2-7b-SAE'.

## G  MORE IMPLEMENTATION DETAILS

To ensure reproducibility, we fix the random seed to 2024 for all experiments. The layer configurations used in our main experiments are as follows: for LLaMA-2-7B-Chat, we use layers 22, 23, 24, 25, and 26; for LLaMA-3.1-8B-Instruct, we utilize layers 3, 12, 18, 20, and 31; for LLaMA-2-13B-Chat, we select layers 5, 12, 24, 32, and 37; and for Qwen-2.5-7B-Instruct, we use layers 19, 21, 23, 24, and 27. The selection criterion—the graph of per-layer AUC scores—is provided in Appendix J.2. Training was conducted with LLaMA-3.1-8B-Instruct on an NVIDIA RTX A6000 48GB GPU, requiring less than 25 minutes per layer.

For the LLMLingua setting, we use LLaMA-2-7B (Touvron et al., 2023)[9] to compute perplexity and apply LongLLMLingua (Jiang et al., 2023b) as the compressor with a compression ratio of 0.55. Selective-Context uses GPT-2 (Radford et al., 2019)[10] to measure perplexity, with a compression ratio of 0.5.

The CAD follows the objective function:

$$\log p_\theta(a_i \mid a_{<i}, q, R) - \alpha \log p_\theta(a_i \mid a_{<i}, q), \tag{11}$$

where $p_\theta(a_i \mid a_{<i}, q, R)$ denotes the probability of generating token $a_i$ conditioned on the retrieved documents $R = M_D(q)$, and $p_\theta(a_i \mid a_{<i}, q)$ is the probability based solely on the internal language

---

[9]meta-llama/Llama-2-7b-hf
[10]openai-community/gpt2

Table 8: Prompt template used for Opin. This template encourages the model to answer the question from the perspective of a fictional character "Bob," using retrieved passages attributed to Bob as supporting context.

---

[SYSTEM PROMPT]
Answer the question based on the given passage. Only give
me the answer and do not output any other words. Bob said,
{retrieval passages}

[RETRIEVAL PASSAGES PROMPT]
Doc 1 (Title: {title}) {content}
Doc 2 (Title: {title}) {content}
Doc 3 (Title: {title}) {content}
Doc 4 (Title: {title}) {content}
Doc 5 (Title: {title}) {content}

[USER PROMPT]
Question: {question} in Bob's opinion?
Answer:

---

Table 9: Performance of CAD with different values of the weighting parameter $\alpha$ across six QA datasets.

| Method | $\alpha$ | NQ | | TriviaQA | | PopQA | | SQuAD | | HotpotQA | | 2WIKI | | AVG |
|--------|----------|-------|-------|----------|-------|-------|-------|-------|-------|----------|-------|-------|-------|-------|
| | | EM | F1 | EM | F1 | EM | F1 | EM | F1 | EM | F1 | EM | F1 | |
| CAD | 1 | **36.67** | **49.60** | **57.67** | **68.47** | **38.83** | **48.22** | **28.50** | 40.41 | **28.33** | **39.13** | 16.83 | 26.99 | **39.97** |
| | 0.5 | 34.33 | 48.59 | 57.50 | 68.34 | 38.67 | 48.05 | 28.33 | **40.69** | 27.50 | 38.69 | **18.00** | **27.48** | 39.68 |

model without external context. The hyperparameter $\alpha$ controls the extent to which the influence of the internal model is penalized. Table 9 presents the performance differences across values of $\alpha$. The best average performance is achieved when $\alpha = 1$, which we adopt as the default configuration in this work.

## H PROMPT TEMPLATES

This section presents the full prompt templates used throughout our experiments. We primarily adopt the prompt format proposed in FlashRAG (Jin et al., 2025). Below, we detail the prompts used for each method:

- **No Retrieval**: See Table 6
- **With Retrieval**, **LLMLingua**, **Selective-Context**, **ITI**, **TruthX**, **CAD**, **IRCAN**, **SPARE**, and **CARE**: See Table 7
- **Opin**: See Table 8
- **Self-RAG**, **SURE**, and **CtrlA**: We use the prompts provided in the original paper.

We observe that the LLaMA-2-13B-Chat (Touvron et al., 2023) model tends to generate additional explanations beyond the desired answer. To mitigate this behavior, we append the instruction "Only give me the answer and do not output any other words." at the end of the user prompt for this model.

## I MORE EXPERIMENTAL RESULTS

### I.1 ROBUSTNESS TO RANDOM SEEDS

To assess the robustness of our method with respect to random seeds, we repeat all experiments with three different random seeds (7, 42, 2024) and report the mean $\pm$ standard deviation over these runs in Table 10. Across datasets and metrics, the variance is relatively small, indicating that the behavior of all methods is reasonably stable under different seeds. In terms of overall performance, CARE achieves an average score of $42.44 \pm 0.81$, consistently outperforming With Retrieval ($40.85 \pm 0.80$). These results demonstrate that the gains brought by CARE are not tied to a particular random seed,

Table 10: Performance with different random seeds (mean $\pm$ standard deviation).

| | NQ | | TriviaQA | | PopQA | |
|---|---|---|---|---|---|---|
| Method | EM | F1 | EM | F1 | EM | F1 |
| No Retrieval | $24.28_{\pm 1.70}$ | $35.27_{\pm 1.60}$ | $55.22_{\pm 2.19}$ | $62.74_{\pm 1.83}$ | $21.83_{\pm 1.66}$ | $28.13_{\pm 0.77}$ |
| With Retrieval | $36.94_{\pm 1.04}$ | $49.98_{\pm 0.73}$ | $61.78_{\pm 2.38}$ | $70.42_{\pm 1.74}$ | $40.50_{\pm 1.06}$ | $49.90_{\pm 1.10}$ |
| SURE | $37.39_{\pm 0.87}$ | $48.71_{\pm 1.77}$ | $56.06_{\pm 5.72}$ | $64.85_{\pm 1.87}$ | $43.22_{\pm 1.22}$ | $48.44_{\pm 0.33}$ |
| IRCAN | $36.50_{\pm 2.37}$ | $48.53_{\pm 1.94}$ | $51.56_{\pm 1.03}$ | $62.27_{\pm 1.38}$ | $37.56_{\pm 1.36}$ | $45.83_{\pm 0.69}$ |
| **CARE** | $\mathbf{40.55}_{\pm 1.77}$ | $\mathbf{52.47}_{\pm 1.35}$ | $\mathbf{62.50}_{\pm 2.09}$ | $\mathbf{70.92}_{\pm 1.34}$ | $\mathbf{43.22}_{\pm 2.11}$ | $\mathbf{50.60}_{\pm 1.26}$ |
| | **SQuAD** | | **HotpotQA** | | **2WIKI** | |
| Method | EM | F1 | EM | F1 | EM | F1 |
| No Retrieval | $12.39_{\pm 0.67}$ | $22.59_{\pm 1.06}$ | $17.83_{\pm 0.85}$ | $26.88_{\pm 0.91}$ | $16.28_{\pm 0.52}$ | $25.73_{\pm 1.53}$ |
| With Retrieval | $31.44_{\pm 1.56}$ | $43.37_{\pm 2.29}$ | $28.17_{\pm 0.94}$ | $38.42_{\pm 0.80}$ | $14.33_{\pm 0.41}$ | $24.90_{\pm 1.15}$ |
| SURE | $27.00_{\pm 2.00}$ | $38.45_{\pm 1.52}$ | $21.56_{\pm 1.70}$ | $30.22_{\pm 2.25}$ | $6.94_{\pm 1.47}$ | $10.59_{\pm 3.47}$ |
| IRCAN | $21.56_{\pm 0.34}$ | $36.84_{\pm 0.96}$ | $23.72_{\pm 0.78}$ | $33.55_{\pm 1.17}$ | $9.78_{\pm 0.48}$ | $20.55_{\pm 0.59}$ |
| **CARE** | $\mathbf{33.67}_{\pm 2.46}$ | $\mathbf{44.71}_{\pm 2.13}$ | $\mathbf{29.05}_{\pm 0.63}$ | $\mathbf{39.36}_{\pm 0.72}$ | $\mathbf{17.72}_{\pm 0.64}$ | $\mathbf{26.86}_{\pm 1.09}$ |

Table 11: Performance of `CARE` on specialized domains. PubMedQA is a benchmark in the medical domain, and LegalBench is a benchmark in the legal domain.

| Method | PubMedQA | LegalBench |
|---|---|---|
| No Retrieval | 46.67 | **72.26** |
| With Retrieval | 55.00 | 71.07 |
| `CARE` | **55.56** | 71.97 |

but instead constitute a reliably reproducible improvement over existing retrieval-augmented QA baselines.

## I.2 TRANSFER TO SPECIALIZED DOMAINS

To examine whether the conflict-aware directions learned from a single QA corpus (NQ) extend to domains requiring specialized expertise, we additionally evaluate CARE on PubMedQA (Jin et al., 2019) and LegalBench (Guha et al., 2023) without any domain-specific fine-tuning. PubMedQA is a biomedical research QA dataset whose possible answers are yes/no/maybe, reflecting the veracity of question statements grounded in scientific literature. We conduct evaluation under the MedRAG (Xiong et al., 2024) framework, retrieving from the MedCorp corpus and reporting the accuracy of model predictions. LegalBench comprises a variety of legal reasoning tasks. It is evaluated within the LRAGE (Minhu Park & Hwang, 2025) framework, which performs retrieval over Pile-of-Law, a legal source provided therein. In these experiments, we utilize BM25 and LLaMA-3.1-8B-Instruct to retrieve the top 5 documents.

Table 11 shows that `CARE` improves PubMedQA performance from 55.00 with simple retrieval to 55.56. This indicates that the intervention of `CARE` not only preserves the benefits of relevant evidence but also enhances its effectiveness. However, on LegalBench, retrieval degrades the base model from 72.26 to 71.07, suggesting that domain mismatch or noisy passages cause harmful conflicts with the model's internal legal knowledge. In this setting, CARE recovers most of the lost accuracy despite using the same retriever. This demonstrates that it can selectively suppress conflict-inducing activations rather than uniformly trusting retrieved content. These improvements imply that the internal features of knowledge conflicts captured by CARE correspond to domain-invariant patterns that transfer to unseen specialized domains, rather than dataset-specific artifacts.

## I.3 EFFECT ON LARGE REASONING MODELS

To assess whether `CARE` also benefits models explicitly designed for multi-step reasoning, we further evaluate it on Qwen3-4B-Thinking (Yang et al., 2025)[11], a representative Large Reasoning Model (LRM). Table 12 reports results on six open-domain QA benchmarks. Adding `CARE` raises the average score to 32.82. These results indicate that `CARE` is complementary to the built-in reasoning capabilities of LRMs, enabling them to use retrieved evidence more reliably rather than naively trusting it.

---

[11] We use the 'Qwen/Qwen3-4B-Thinking-2507' checkpoint released on HuggingFace.

Table 12: Performance on Qwen3-4B-Thinking, a Large Reasoning Model.

| Method | NQ F1 | NQ ACC | TriviaQA F1 | TriviaQA ACC | PopQA F1 | PopQA ACC | SQuAD F1 | SQuAD ACC | HotpotQA F1 | HotpotQA ACC | 2WIKI F1 | 2WIKI ACC | AVG |
|---|---|---|---|---|---|---|---|---|---|---|---|---|---|
| No Retrieval | 15.95 | 13.33 | 7.17 | 21.67 | 4.67 | 20.00 | 14.32 | 11.67 | 11.04 | 20.00 | 6.57 | 26.67 | 14.42 |
| With Retrieval | 18.26 | 40.00 | 25.50 | 46.67 | 18.44 | 38.33 | 21.56 | **21.67** | **20.60** | 28.33 | 8.21 | 26.67 | 26.19 |
| CARE | **23.60** | **50.00** | **40.07** | **56.67** | **29.38** | **56.67** | **22.69** | **21.67** | 20.26 | **30.00** | **14.48** | **28.33** | **32.82** |

Table 13: Comparison of average retrieval, inference, and total latency (in seconds) across different methods.

| Method | Retrieval Time | Inference Time | Total Time |
|---|---|---|---|
| No Retrieval | - | 0.3585 | 0.3585 |
| With Retrieval | 1.4054 | 0.4677 | 1.8731 |
| LLMLingua | 2.1858 | 0.4266 | 2.6124 |
| Selective-Context | 1.6948 | 0.4430 | 2.1378 |
| SURE | 1.4054 | 6.0109 | 7.4163 |
| Opin | 1.4054 | 0.5264 | 1.9318 |
| CAD | 1.4054 | 1.8400 | 3.2454 |
| IRCAN | 1.3755 | 1.0695 | 2.4450 |
| CARE | 1.4054 | 0.4232 | 1.8286 |

# J   MORE ANALYSIS

## J.1   ANALYSIS OF CARE DESIGN CHOICES

In this section, we investigate how the key design choices of CARE affect its performance. Table 14 presents ablation results on (i) the autoencoder latent dimension, (ii) the use of nonlinearity in the autoencoder, and (iii) the choice of layers where the intervention is applied.

The autoencoder in CARE requires an appropriately compressed latent space. We use one quarter of the hidden size ($d/4$) as the latent dimension and compare it to a setting where the latent dimension is increased to $d/2$. The results show that this larger-latent setting consistently degrades overall performance. A plausible explanation is that an overly large latent space preserves complex signals and noise that are not directly related to knowledge conflict, thereby introducing interference during conflict mitigation. By contrast, the stronger compression at $d/4$ effectively filters out such unnecessary factors and retains only the information needed to distinguish conflict states.

The encoder/decoder MLPs of CARE use the LeakyReLU activation function. When we remove all nonlinearities and replace the autoencoder with a purely linear one, the average performance drops sharply from 41.37 to 18.62. This suggests that simple linear projection is insufficient to capture the latent structure required to distinguish between knowledge-conflict states. That nonlinearity plays a crucial role in isolating and correcting conflict signals.

For layer selection, we first compute an AUC score for each layer and choose the layer with the highest score for intervention. If multiple layers share the highest score, we randomly select one among them. To validate the robustness of this choice, we compare our original layer selection with a variant that intervenes on the top 5 layers ranked by AUC (layers 18, 20, 28, 29, 31). The difference in average performance between the two settings is only 0.16, and the overall trends are very similar. This suggests that what matters is not the highly specialized functional role of a particular layer, but rather that each chosen layer provides a strong signal for distinguishing between conflict and non-conflict states.

## J.2   LAYER-WISE AUC OF LATENT SPACE SEPARATION

To analyze how well the latent representations are separated at each layer, we compute AUC for latent space separation across transformer layers. Figure 5 visualizes the AUC values measured at each layer. In the early layers, the AUC scores are relatively low and unstable, indicating limited class separation. However, up to approximately the 12th layer, AUC scores steadily increase, suggesting improved separability of representations. After the 12th layer, AUC values generally remain above 0.9, reflecting stable and high separability. This suggests that representations in the middle-to-late layers of the model contain richer information necessary for separating classes. These layers are

Table 14: Ablation study on CARE design choices. **Latent Dimension** $d/2$ denotes a setting where the autoencoder latent dimension is expanded to $d/2$ instead of the default, **Without LeakyReLU** denotes a purely linear autoencoder without any nonlinear activations, and **Upper 5 Layers** denotes a setting where interventions are applied only to the top 5 layers instead of the original layer configuration.

| Method | NQ | | TriviaQA | | PopQA | | SQuAD | | HotpotQA | | 2WIKI | | AVG |
|---|---|---|---|---|---|---|---|---|---|---|---|---|---|
| | EM | F1 | EM | F1 | EM | F1 | EM | F1 | EM | F1 | EM | F1 | |
| CARE | 38.50 | 51.31 | 59.67 | 69.96 | 40.33 | 48.84 | 30.33 | 41.85 | 29.50 | 40.36 | 18.33 | 27.51 | 41.37 |
| Latent Dimension $d/2$ | 33.16 | 47.64 | 51.50 | 63.39 | 34.00 | 44.21 | 23.83 | 34.70 | 20.67 | 33.01 | 11.00 | 18.95 | 34.67 |
| Without LeakyReLU | 8.33 | 20.55 | 25.00 | 40.07 | 11.66 | 27.65 | 8.33 | 18.39 | 16.67 | 22.32 | 10.00 | 14.48 | 18.62 |
| Upper 5 Layers | 39.50 | 51.32 | 59.83 | 69.67 | 41.16 | 48.95 | 30.66 | 41.79 | 30.33 | 40.10 | 17.83 | 27.22 | 41.53 |

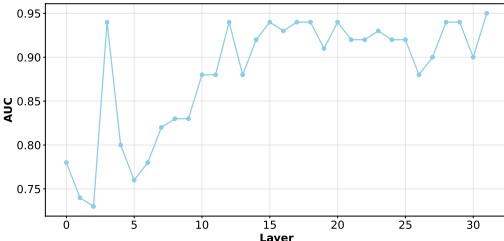

Figure 5: Layer-wise AUC scores for latent space separation. Higher AUC indicates better discriminative capability of the representations at each transformer layer.

known to encode high-level semantic features and factuality-related signals Chuang et al. (2024); Wu et al. (2025); Skean et al. (2025), and are believed to play a central role in integrating information under conflicting conditions. Our findings are consistent with prior empirical studies Chen et al. (2024); Seo et al. (2025), which report that layers most important for hallucination detection tend to be concentrated in the middle-to-late parts of the network.

### J.3    TOKEN-WISE VISUALIZATION

To more precisely analyze how the learned latent space differs within model responses, we conduct a token-level analysis based on representations. Specifically, we extract the representation for each token and compute a score based on its distance to pre-defined positive and negative centroids. A higher score indicates that the token representation is closer to the positive centroid, suggesting a stronger alignment with knowledge-alignment information. Figure 6 shows heatmaps of the token-wise scores for a positive and a negative response. In the positive case, the scores remain relatively high and consistent throughout the response. In contrast, the negative sample exhibits lower and more irregular scores, indicating less alignment with reliable information. This token-level visualization provides intuitive insight into where knowledge conflicts may arise, allowing for fine-grained interpretability of model behavior under our training method.

### J.4    VALIDATION OF NEGATIVE SAMPLE DISTRIBUTION

We ablate the composition of negative answers by varying the proportion of Parametric vs. Noise negatives while holding all other settings fixed. Concretely, we compare three configurations: a balanced mix (50%/50%), parametric-only (100%/0%), and noise-only (0%/100%). As summarized in Table 15, the balanced configuration achieves the best average performance, outperforming both parametric-only and noise-only. At the dataset level, a similar trend holds. Meanwhile, the noise-only setting attains the best F1 on TriviaQA and HotpotQA, and slightly higher 2WIKI. These results suggest that the two negative types provide complementary supervision signals: parametric negatives better regularize entity-centric and short-form QA, whereas noise negatives sharpen robustness to retrieval noise and multi-hop reasoning. If CARE were trained only along a single truthfulness dimension, such a complementary effect would not arise. This supports our claim that it is designed to handle two forms of conflict. Overall, the balanced mixture confers the most consistent gains across benchmarks, indicating that aligning the representation against both parametric confounders and retrieval-induced noise is crucial for generalization in RAG.

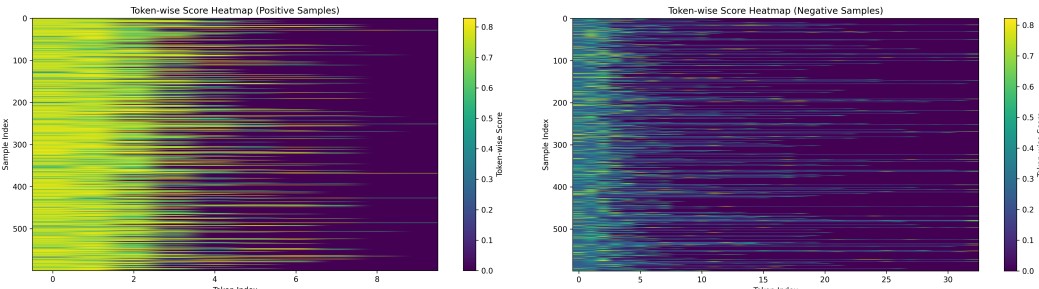

Figure 6: Token-wise score heatmaps for a positive (left) and a negative (right) response. Scores reflect proximity to a positive latent centroid, with higher values indicating stronger alignment with factual content.

Table 15: Performance versus changes in the negative-sample distribution. Balanced denotes an equal mixture of parametric and noise samples, while Parametric-only and Noise-only indicate distributions consisting exclusively of parametric or noise negatives, respectively.

| Setting | NQ | | TriviaQA | | PopQA | | SQuAD | | HotpotQA | | 2WIKI | | AVG |
|---|---|---|---|---|---|---|---|---|---|---|---|---|---|
| | EM | F1 | EM | F1 | EM | F1 | EM | F1 | EM | F1 | EM | F1 | |
| Balanced (50% / 50%) | **38.50** | **51.31** | **59.67** | 69.96 | **40.33** | **48.84** | **30.33** | **41.85** | **29.50** | 40.36 | **18.33** | **27.51** | **41.37** |
| Parametric-only (100% / 0%) | 35.83 | 50.14 | 58.33 | 70.00 | 37.83 | 47.72 | 29.00 | 41.46 | 28.17 | 40.63 | 17.83 | 27.48 | 40.37 |
| Noise-only (0% / 100%) | 36.00 | 50.09 | 58.67 | **70.48** | 37.50 | 47.69 | 29.00 | 41.48 | 28.67 | **40.86** | **18.33** | 27.68 | 40.54 |

## J.5 EFFECT OF APPLYING CAD

We observe in the original IRCAN paper that combining CAD markedly mitigates conflicts. Accordingly, as shown in Table 16, we empirically evaluate the performance of attaching CAD to both IRCAN and CARE. As a result, integrating CAD at the decoding stage yields a significant improvement in overall accuracy. Compared to CARE, CARE-CAD achieves an average absolute gain of 1.27, surpassing the prior methods IRCAN and IRCAN-CAD by 7.13 and 4.65, respectively. Across datasets, performance generally improves on all benchmarks, confirming the beneficial effect of introducing CAD. However, adding CAD entails a trade-off of increased inference time.

## K CASE STUDY

To better understand model behavior under challenging queries, Table 17 presents two representative cases illustrating how different methods respond in the presence of internal and external knowledge conflicts. In the first example, confusion arises between internal knowledge (Rab C. Nesbitt) and retrieved documents that frequently mention the boat name "The Vital Spark." Some models return hallucinated or misleading answers such as The Vital Spark or Rab C. Nesbitt. In contrast, CARE correctly identifies Para Handy by integrating two key pieces of textual evidence: "The Tales of Para Handy, starring Gregor Fisher" and "captain of the puffer 'Vital Spark'," effectively resolving the ambiguity. The second example focuses on confusion induced by external knowledge. While some models misinterpret the acronym "COBRA" and respond with Room or Civil, CARE successfully identifies Cabinet by reasoning over the retrieved phrase "Cabinet Office Briefing Rooms", which explicitly defines the acronym. These cases demonstrate that CARE is robust against knowledge conflicts and can reliably extract answers grounded in retrieved evidence, even under noisy or misleading conditions.

Table 16: Performance with CAD additionally applied to IRCAN and CARE.

| Method | NQ | | TriviaQA | | PopQA | | SQuAD | | HotpotQA | | 2WIKI | | AVG |
|--------|----|----|----|----|----|----|----|----|----|----|----|----|-----|
| | EM | F1 | EM | F1 | EM | F1 | EM | F1 | EM | F1 | EM | F1 | |
| IRCAN | 37.83 | 49.55 | 50.67 | 61.56 | 38.17 | 45.82 | 21.17 | 35.67 | 23.17 | 33.65 | 9.17 | 19.71 | 35.51 |
| IRCAN-CAD | 41.17 | 51.36 | 57.17 | 65.35 | 41.00 | 46.99 | 26.33 | 36.59 | 24.50 | 32.59 | 13.00 | 19.77 | 37.99 |
| CARE | 38.50 | 51.31 | 59.67 | **69.96** | 40.33 | 48.84 | 30.33 | 41.85 | 29.50 | **40.36** | 18.33 | **27.51** | 41.37 |
| CARE-CAD | **43.00** | **53.39** | **61.50** | 69.88 | **44.50** | **50.81** | **31.83** | **41.88** | **29.83** | 39.29 | **18.67** | 27.14 | **42.64** |

Table 17: Case study on two challenging examples involving knowledge conflict.

| | |
|---|---|
| **Question 1** | **Played on TV by Gregor Fisher, whose boat was called The Vital Spark?** |
| Document 1 | (Title: The Vital Spark) The Vital Spark was a BBC Scotland television series set in the western isles of Scotland in the 1930s, based on the Para Handy books by Neil Munro. It starred Roddy McMillan as Peter... |
| Document 2 | (Title: The Tales of Para Handy) The series followed the "Vital Spark's" adventures around the coastal waters of west Scotland and the various schemes that Para Handy would get himself and his crew involved in... |
| Document 3 | (Title: Para Handy - Master Mariner) several dramatizations of the "Para Handy" stories, being followed by "The Vital Spark" in 1965-6 (remade in 1973-4) starring Roddy McMillan in the title role, and by **"The Tales of Para Handy" in 1994-5, starring Gregor Fisher.** Season 1 Para Handy - Master Mariner Para Handy - Master Mariner is a Scottish television series set in the western isles of Scotland in the 1930s, based on the Para Handy books by Neil Munro. It starred Duncan Macrae as **Peter "Para Handy" MacFarlane, captain of the puffer "Vital Spark"**... |
| Document 4 | (Title: Vital Spark) Gregor Fisher in the lead role alongside Sean Scanlan as Dougie, Andrew Fairlie as Sunny Jim and Rikki Fulton as Dan McPhail. This ran for two series, a total of nine episodes, in 1994 and 1995... |
| Document 5 | (Title: Vital Spark) Master Mariner, which ran from 1959–60, starred Duncan Macrae (Para Handy), Roddy McMillan (The Mate), and John Grieve (Dan MacPhail, the engineer). Six episodes were made... |
| Ground-truth Answer | Para Handy |
| No Retrieval | Rab C. Nesbitt. ✗ |
| With Retrieval | The Vital Spark ✗ |
| SURE | Para Handy ✓ |
| IRCAN | The Vital Spark ✗ |
| **CARE** | **Para Handy ✓** |
| **Question 2** | **In the context of UK government, for what does the C stand in the acronym COBRA?** |
| Document 1 | (Title: Civil Contingencies Secretariat) Civil Contingencies Committee, often informally referred to as COBRA from the name of the room used, is a forum for ministers and senior officials to discuss... |
| Document 2 | (Title: Cabinet Office Briefing Rooms) committee after the 7 July 2005 London bombings and at other intervals from 2005 to 2007, was highly critical of its workings in his book "The Terrorist Hunters". A single photo of one of the rooms in COBR was released in 2010 in response to a Freedom of Information Act request. Cabinet Office Briefing Rooms The **Cabinet Office Briefing Rooms (COBR, often mistakenly referred to as COBRA)** are a group of meeting rooms in the Cabinet Office at 70 Whitehall, often used for different committees which co-ordinate the actions of bodies within **the Government of the United Kingdom** in response... |
| Document 3 | (Title: Cabinet Office Briefing Rooms) **The Cabinet Office Briefing Rooms (COBR, often mistakenly referred to as COBRA)** are a group of meeting rooms in the Cabinet Office at 70 Whitehall, often... |
| Document 4 | (Title: The Cobra (novel)) The Cobra is a 2010 thriller novel by Frederick Forsyth about the international cocaine trade. In it, an unnamed Obama-like (said to have a wife named 'Michelle' and... |
| Document 5 | (Title: COBRA (avant-garde movement)) COBRA (or CoBrA) was a European avant-garde movement active from 1948 to 1951. The name was coined in 1948 by Christian Dotremont from the initials of... |
| Ground-truth Answer | Cabinet |
| No Retrieval | Cabinet ✓ |
| With Retrieval | There is no clear answer in the given passages. ✗ |
| SURE | Civil ✗ |
| IRCAN | Room ✗ |
| **CARE** | **Cabinet ✓** |

