# OpenReview forum: "Conflict-Aware Representation Editing for Robust Retrieval-Augmented Generation"
_ICLR.cc/2026/Conference — Submitted to ICLR 2026_

### Official Review · Reviewer_nr8n · 2025-10-25

**Soundness:** 3
**Presentation:** 3
**Contribution:** 4
**Rating:** 6
**Confidence:** 4

**Summary:**

This paper introduces **conflict-aware representation editing (CARE)**, a representation-editing-based method to address the challenge of knowledge conflict in retrieval-augmented generation (RAG).

CARE is based on autoencoders and resolve knowledge conflicts by (1) carefully curating training data consisting of positive answers and negative answers (including parametric and noise answers) (2) editing in the autoencoder latent space.
Data curation allows the autoencoder to distinguish between knowledge aligned/conflicted answers, while latent space editing shifts model representations towards the conflict-free direction.

Experiment results validate that CARE outperforms RAG baselines, conflict resolution methods and truthfulness representation editing methods on six standard and multi-hop QA tasks.
Further experiments including ablation study, hyperparameter sensitivity and visualization support that CARE is a robust enhancement to RAG.

**Strengths:**

1. Originality

   The paper is novel by using representation editing to solve the knowledge conflict challenges of RAG.
   This approach is superior to RAG-enhancement methods and conflict resolutions methods in terms of QA performance and efficiency, outperforms truthfulness representation editing methods (e.g. ITI) in the setting of RAG, and it is better suited for RAG tasks than knowledge editing methods for its ability to perform on-the-fly knowledge adjustment.
2. Quality

   The paper is technically sound.
   The method is clearly explained, experiments are well-designed and results are convincing.
3. Clarity

   Overall, this paper is well written by clearly communicating the motivation, key insight, and how it contributes to the field.
4. Significance

   First, this work contributes to the field of RAG.
   It shows that representation editing is both effective and efficient when compared with RAG methods like SURE and conflict resolution methods such as IRCAN. This work could lead to a paradigm shift towards more robust RAG.

   Second, this work is also meaningful to the field of representation editing.
   It shows concrete evidence that representation editing could be effective in long-context and knowledge-extensive QA tasks.

**Weaknesses:**

(Authors do **not** need to refer to points raised in this section since the main points are already mentioned in *"Questions" section*.)
1. W1: Reproducibility

   The authors have provided extensive details in the paper that could facilitate reproduction of main results. However, code implementations are not provided either as supplementary material or present in an anonymous link.
2. W2: Discussions on potential limitations

   The paper does not thoroughly discuss limitations of this work.
3. W3: Related work

   The paper does not take into account broader literature in mechanistic interpretability, particularly studies on sparse dictionary learning that train sparse autoencoders to reconstruct model representations (details in "Questions" section).

**Questions:**

**Major questions (that could affect rating)**
1. **Question 1**: Reproducibility

   Do authors have plans of open-sourcing their code? If so, it would be helpful to the research community; if not, please elaborate since it may adversely impact the paper's contribution to the field.
2. **Question 2**: Explanation for suboptimal performance on PopQA in Table 1

   Table 1 shows that although CARE generally outperforms most baseline methods, it consistently fails to yield optimal performance on PopQA. What might be the reason for this performance gap? Is it due to certain intrinsic characteristics of the PopQA task, or does it stem from the method CARE itself?
3. **Question 3**: Discussions on potential limitations

   The paper could expand on potential limitations, which would show the authors' in-depth insights in this field and would be meaningful in inspiring future work.
4. **Question 4**: Related work

   The paper does not take into account broader literature in mechanistic interpretability, particularly those on sparse dictionary learning (SDL)[1-3]. These works are related to this paper since autoencoders are trained to reconstruct model representations.
   [1-3] are different from the method of CARE since [1-3] use sparse autoencoders, use different auxiliary losses and have much larger latent space sizes.

   Nevertheless, incorporating SDL into the picture could help the paper set a broader context for potential readers, especially considering that this work touches on knowledge-related mechanisms in representation space (lines 82-85).
5. **Question 5**: Is CARE resolving knowledge conflicts or eliciting truthful answers?

   Truthfulness representation editing methods such as ITI and TruthX are introduced as baselines. Despite the narrative of this paper that CARE is designed to solve knowledge conflicts, it *might* be the case that CARE is eliciting truthful answers.

   Therefore this question is a conceptual and fundamental one: Is CARE directly improving truthfulness, or is it indirectly improving truthfulness by resolving knowledge conflicts?

   Could current evidence of Table 4 (Analysis of knowledge conflict resolution with external knowledge) help answer this question?
   Furthermore, could this question be answered by decomposing the negative sample distribution of Figure 4 into two distributions, one for parametric negative answers and another for noise negative answers?
6. **Question 6**: Effect of latent space dimension on performance

   The paper has shown ablation studies with respect to training loss functions and directional coefficients.
   Beyond these aspects, additional information regarding the latent space dimension could be useful in terms of the trade-off between computational efficiency and task performance.
   This concern is motivated by previous works on linear representation steering where 1D linear vectors are sufficient for good task performance[4-6].


**Minor questions and suggestions (that are not considered to affect rating)**
1. **Minor question 1**: GPU model specs

   It is mentioned multiple times in the paper that "A6000 40GB" GPUs are used. However, as far as I know the A6000 model only has the 48GB configuration. Therefore I am confused if this is simply a typo.
2. **Minor question 2**: MLP non-linear activation function

   What is the activation function used for encoder/decoder MLPs (ReLU/GeLU/...)?
   This information is useful since I am interested in *how much non-linearity* in introduced into the CARE autoencoder, which could be expressed as the following question: can CARE achieve the same level of performance with an entirely linear autoencoder?
3. **Minor question 3**: Inconsistency between text and figure.

   The paper states at lines 202-204 that the dimension of the autoencoder latent space is *smaller* than representation space.
   However, in Figure 1, the latent space is *larger* representation space, judging by the shapes of the encoder and decoder.
   Therefore authors might need to modify the figure to help with understanding.
4. **Suggestion 1**: Efficiency-related results and discussions

   Efficiency of RAG enhancements is a critical concern. The authors have done a good job by including efficiency analysis in Section G.3, showing that CARE introduces negligible latency with respect to the "With Retrieval" baseline.
   This result is important and deserves mentioning in the main body to highlight efficiency advantages.
5. **Suggestion 2**: Variance and statistical significance

   Experiment results could be enhanced with variance across seeds and statistical significance. These information could shed light on whether CARE is stable across training runs.


References:

[1] Towards monosemanticity: Decomposing language models with dictionary learning. (2023)
[2] Sparse autoencoders find highly interpretable features in language models. (ICLR 2024)
[3] Scaling and evaluating sparse autoencoders. (ICLR 2025)
[4] Inference-Time Intervention: Eliciting Truthful Answers from a Language Model (NeurIPS 2023)
[5] Steering llama 2 via contrastive activation addition. (ACL 2024)
[6] Activation addition: Steering language models without optimization. (2023)

---

> ### Author Response · Authors · 2025-11-21
> **Response to Reviewer nr8n (1)**
>
> Thank you for your thoughtful comments and for taking the time to review our paper. We hope our replies and revisions sufficiently address your concerns and enhance clarity. Additionally, all modifications to the manuscript have been highlighted in blue for easy reference. Please do not hesitate to reach out if you have any further questions or feedback.
>
> **Q1.** Reproducibility
>
> > We are committed to releasing our code in the near future.
> >
>
> **Q2.** Explanation for suboptimal performance on PopQA in Table 1
>
> > We have observed consistent improvements in PopQA F1 for CARE across various models. However, we acknowledge that we did not sufficiently discuss the case where CARE underperforms other methods on PopQA for LLaMA-2-13B-Chat, and we have revised the manuscript to address this in the Limitations section.
> >
> >
> > PopQA primarily consists of relatively simple questions about commonsense and factual knowledge. In other words, the accuracy of the model’s parametric knowledge is especially critical in this dataset. Table 10 shows that training CARE using only parametric-error negatives leads to better performance on PopQA than using only retrieval-noise negatives. Because PopQA is particularly sensitive to parametric errors, this balanced setting may limit the potential gains for certain models. This outcome can be interpreted as a consequence of our design choice to address a broader range of retrieval-induced conflicts rather than focusing solely on parametric errors.
> >
>
> **Q3.** Discussions on potential limitations
>
> > This work has several limitations that should be addressed in future research. We clearly articulate these points in a dedicated Limitations section.
> >
> > - Sensitivity to the training data distribution: Our method is primarily trained on datasets where retrieval-induced confusion within long contexts is the dominant source of errors. As a result, performance gains may be less pronounced on datasets such as PopQA, where errors stem more from straightforward factual verification and parametric inaccuracies. Future work should investigate training strategies that adjust the mixture of negative samples according to the characteristics of the target domain.
> > - Potential risks of global linear editing: CARE applies a single, pre-learned direction vector uniformly to all inputs during inference. While our experiments show that this global intervention is generally safe on average, it still carries potential risks—particularly when the model already possesses correct parametric knowledge, where unnecessary perturbation might distort correct answers. Future extensions could incorporate adaptive mechanisms that estimate the degree of conflict in real time and dynamically modulate the editing strength accordingly.
> > - Empirical dependence in hyperparameter selection: The choice of editing layers and the editing strength currently relies on empirical procedures such as AUC-based layer scoring and grid search. Further research is needed to develop automated methods that can predict or determine optimal intervention points across diverse model architectures.
>
> **Q4.** Related work
>
> > We have incorporated the works recommended by the reviewer and added the following discussion to the Related Work section.
> >
> >
> > **Mechanistic Interpretability.** Recent mechanistic interpretability work has leveraged sparse dictionary learning (SDL) and Sparse Autoencoders (SAEs) to decompose internal LLM representations into interpretable, monosemantic features [1–3]. These approaches typically rely on overcomplete latent spaces and strong sparsity regularization to disentangle complex activation patterns. Building on this line of work, [4] further demonstrated that SAE-derived latent features can be directly manipulated to steer knowledge-selection behaviors in LLMs, highlighting the controllability and interpretability enabled by sparse latent structures. Although our method also employs an autoencoder architecture to analyze representation space, its purpose differs fundamentally from that of SAEs. Instead of pursuing unsupervised feature decomposition for interpretability, we utilize a compressed, dense latent space trained with supervised contrastive signals. This allows us to identify and manipulate global directions tied to knowledge conflict, prioritizing robustness and controllability of model behavior.
> >
> > [1] Bricken et al. Towards monosemanticity: Decomposing language models with dictionary learning. (2023).
> >
> > [2] Hoagy et al. Sparse autoencoders find highly interpretable features in language models. ICLR (2024).
> >
> > [3] Gao et al. Scaling and evaluating sparse autoencoders. ICLR (2025).
> >
> > [4] Zhao et al. Steering Knowledge Selection Behaviours in LLMs via SAE-based Representation Engineering. NAACL (2024).
> >

---

> ### Author Response · Authors · 2025-11-21
> **Response to Reviewer nr8n (2)**
>
> **Q5.** Is CARE resolving knowledge conflicts or eliciting truthful answers?
>
> > CARE improves truthfulness **indirectly** by resolving knowledge conflicts. The goal of this work is not to learn the abstract notion of truthfulness itself, but to address the **concrete problem of conflict** between internal parametric knowledge and external information.
> >
> > - **Knowledge Conflict Resolution Analysis** (Table 4): Table 4 evaluates model performance from the perspective of conflict rather than truthfulness. CARE achieves the highest Resolution score (36.42), indicating that it is most effective at correcting incorrect internal knowledge using external evidence. CARE also obtains the lowest Conflict score (6.13), meaning it is the most robust at preserving correct prior knowledge without being misled by noise or contradictory signals.
> > - **Negative Sample Distribution** (Table 10): Table 10 further examines this distinction.
> >     - Hypothesis 1 (if CARE were a truthfulness editor): If CARE simply learned a single truth vs. falsehood axis, both parametric errors and noise-induced errors would collapse into the same false region.
> >     - Hypothesis 2 (if CARE resolves conflicts): If CARE is designed to resolve conflicts, the model’s own mistakes (parametric errors) and mistakes caused by misleading external information (noise errors) should reflect distinct supervisory signals.
> >
> >     We trained CARE using three negative sample configurations: 100% parametric errors, 100% noise errors, and a balanced 50/50 mixture. The balanced configuration achieved the highest performance, demonstrating that the two error types provide **complementary supervision**. In other words, $Z_{neg-parametric}$ and $Z_{neg-noise}$ correspond to different underlying failure modes, and exposing the model to both is crucial for robustness. If CARE learned only a single truthfulness axis, this complementary effect would not appear. This supports our claim that CARE is explicitly designed to handle **multiple forms of conflict**, not to learn truthfulness directly. We have incorporated this discussion into the analysis of Tables 4 and 10.
> >
> >
> > Furthermore, the negative sample distribution in Figure 4 does not separate cleanly into two clusters (parametric negative vs. noise negative) because, although these two error types originate from different causes, they share a substantial amount of common incorrectness signal at the representation level.
> >
>
> **Q6.** Effect of latent space dimension on performance
>
> > The vector we use to actually steer the model’s behavior is one-dimensional, and constructing a latent space that yields a reliable direction requires an optimally compressed latent dimensionality. We set this to one-quarter of the model’s hidden size and compared it with a half-dimensional setting. Our experiments show that latent dimensions larger than $d/4$ in fact degrade performance. When the latent space becomes too large, it tends to preserve unnecessary complex or noisy information, which introduces interference in conflict mitigation. In contrast, optimal compression effectively filters out such distracting factors and maximizes the separability required for robust conflict-aware steering. We clarify this design choice in Section 4.2 and Appendix J.1.
> >
> >
> >
> > |  | NQ (EM) | NQ (F1) | TriviaQA (EM) | TriviaQA (F1) | PopQA (EM) | PopQA (F1) | SQuAD (EM) | SQuAD (F1) | HotpotQA (EM) | HotpotQA (F1) | 2WIKI (EM) | 2WIKI (F1) | AVG |
> > | --- | --- | --- | --- | --- | --- | --- | --- | --- | --- | --- | --- | --- | --- |
> > | CARE | 38.50 | 51.31 | 59.67 | 69.96 | 40.33 | 48.84 | 30.33 | 41.85 | 29.50 | 40.36 | 18.33 | 27.51 | 41.37 |
> > | Vector Dimension d/2 | 33.16 | 47.64 | 51.50 | 63.39 | 34.00 | 44.21 | 23.83 | 34.70 | 20.67 | 33.01 | 11.00 | 18.95 | 34.67 |
>
> **MQ1.** GPU model specs
>
> > Thank you for pointing this out. You are correct that the GPU used in our experiments has 48GB of memory. Specifically, we conducted all experiments using the NVIDIA RTX A6000 (48GB). We appreciate the reviewer’s careful observation, and we have corrected this in the revised manuscript.
> >

---

> ### Author Response · Authors · 2025-11-21
> **Response to Reviewer nr8n (3)**
>
> **MQ2.** MLP non-linear activation function
>
> > In the encoder/decoder MLPs of CARE, we use the LeakyReLU activation function. We removed all non-linearities and repeated the experiments. The results show a substantial performance drop: the average score decreases from 41.37 to 18.62 without LeakyReLU. This clearly indicates that a linear autoencoder is *insufficient* for capturing the latent-space structure needed for effective representation correction. We include this analysis in the Appendix J.1 and discuss the role of nonlinearity in our architecture.
> >
> >
> >
> > |  | NQ (EM) | NQ (F1) | TriviaQA (EM) | TriviaQA (F1) | PopQA (EM) | PopQA (F1) | SQuAD (EM) | SQuAD (F1) | HotpotQA (EM) | HotpotQA (F1) | 2WIKI (EM) | 2WIKI (F1) | AVG |
> > | --- | --- | --- | --- | --- | --- | --- | --- | --- | --- | --- | --- | --- | --- |
> > | CARE | 38.50 | 51.31 | 59.67 | 69.96 | 40.33 | 48.84 | 30.33 | 41.85 | 29.50 | 40.36 | 18.33 | 27.51 | 41.37 |
> > | Without LeakyReLU | 8.33 | 20.55 | 25.00 | 40.07 | 11.66 | 27.65 | 8.33 | 18.39 | 16.67 | 22.32 | 10.00 | 14.48 | 18.62 |
>
> **MQ3.** Inconsistency between text and figure
>
> > We appreciate the reviewer for pointing out this potential source of confusion. We recognize that the original Figure 1 may be misleading, and we have revised it to ensure greater clarity and consistency with the text.
> >
>
> **S1.** Efficiency-related results and discussions
>
> > Thank you for raising the point about efficiency. We have revised the manuscript to explicitly discuss this in Section 5.3 of the main text.
> >
>
> **S2.** Variance and statistical significance
>
> > Following the reviewer’s suggestion, we evaluated our method using three different random seeds (7, 42, 2024) and reported the mean ± standard deviation in the Appendix I.1. Across all metrics, the standard deviation remains small (typically around 1.0 or lower), indicating that the model is stable with respect to random initialization. CARE achieves an overall score of 42.44 ± 0.81, consistently outperforming both the retrieval baseline (40.85 ± 0.80) and the no-retrieval baseline (29.10 ± 0.37) across all seeds. These results confirm that CARE’s improvement is not a single-seed artifact, but a **reliably reproducible enhancement**.
> >
> >
> >
> > |  | NQ (EM) | NQ (F1) | TriviaQA (EM) | TriviaQA (F1) | PopQA (EM) | PopQA (F1) | SQuAD (EM) | SQuAD (F1) | HotpotQA (EM) | HotpotQA (F1) | 2WIKI (EM) | 2WIKI (F1) | AVG |
> > | --- | --- | --- | --- | --- | --- | --- | --- | --- | --- | --- | --- | --- | --- |
> > | No Retrieval | 24.28 ± 1.70 | 35.27 ± 1.60 | 55.22 ± 2.19 | 62.74 ± 1.83 | 21.83 ± 1.66 | 28.13 ± 0.77 | 12.39 ± 0.67 | 22.59 ± 1.06 | 17.83 ± 0.85 | 26.88 ± 0.91 | 16.28 ± 0.52 | 25.73 ± 1.53 | 29.10 ± 0.37 |
> > | With Retrieval | 36.94 ± 1.04 | 49.98 ± 0.73 | 61.78 ± 2.38 | 70.42 ± 1.74 | 40.50 ± 1.06 | 49.90 ± 1.10 | 31.44 ± 1.56 | 43.37 ± 2.29 | 28.17 ± 0.94 | 38.42 ± 0.80 | 14.33 ± 0.41 | 24.90 ± 1.15 | 40.85 ± 0.80 |
> > | SURE | 37.39 ± 0.87 | 48.71 ± 1.77 | 56.06 ± 5.72 | 64.85 ± 1.87 | **43.22** ± 1.22 | 48.44 ± 0.33 | 27.00 ± 2.00 | 38.45 ± 1.52 | 21.56 ± 1.70 | 30.22 ± 2.25 | 6.94 ± 1.47 | 10.59 ± 3.47 | 36.12 ± 0.53 |
> > | IRCAN | 36.50 ± 2.37 | 48.53 ± 1.94 | 51.56 ± 1.03 | 62.27 ± 1.38 | 37.56 ± 1.36 | 45.83 ± 0.69 | 21.56 ± 0.34 | 36.84 ± 0.96 | 23.72 ± 0.78 | 33.55 ± 1.17 | 9.78 ± 0.48 | 20.55 ± 0.59 | 35.68 ± 0.14 |
> > | **CARE** | **40.55** ± 1.77 | **52.47** ± 1.35 | **62.50** ± 2.09 | **70.92** ± 1.34 | **43.22** ± 2.11 | **50.60** ± 1.26 | **33.67** ± 2.46 | **44.71** ± 2.13 | **29.05** ± 0.63 | **39.36** ± 0.72 | **17.72** ± 0.64 | **26.86** ± 1.09 | **42.44** ± 0.81 |
> >

---

### Official Review · Reviewer_Z5kc · 2025-10-30

**Soundness:** 2
**Presentation:** 3
**Contribution:** 2
**Rating:** 4
**Confidence:** 3

**Summary:**

The paper proposes Conflict-Aware Representation Editing (CARE), a method to improve the robustness of Retrieval-Augmented Generation (RAG) systems under knowledge conflicts and noisy retrievals. The central idea is that large language models (LLMs) exhibit distinguishable internal activation patterns when external context contradicts parametric knowledge. CARE introduces an autoencoder trained on positive (correct) and negative (conflicted/noisy) QA pairs to identify such patterns and learns a latent “conflict-mitigation direction.” During inference, it adjusts neuron activations along this learned direction in specific layers to suppress conflict signals and enhance reliability. Experiments across six QA benchmarks (NQ, TriviaQA, PopQA, SQuAD, HotpotQA, 2Wiki) and four LLMs (LLaMA-2/3, Qwen-2.5) show consistent improvements over prior RAG and representation-editing baselines such as CAD, IRCAN, and TruthX.

**Strengths:**

See Summary.

**Weaknesses:**

1.	Conceptual validity of training supervision — The autoencoder is trained using externally labeled correct/incorrect answers rather than the model’s internal conflict cases. This design may not truly capture internal conflict representations, making it unclear why the learned δ direction should correspond to factual reliability.
2.	Ambiguous knowledge-source attribution — The method can identify conflict-related activation patterns but cannot determine which knowledge (parametric or retrieved) is correct. Without this distinction, editing representations might sometimes steer the model toward incorrect sources.
3.	Lack of interpretability and mechanism insight — The paper does not analyze what features the autoencoder encodes, or whether its latent dimensions correspond to known interpretable features, as done in sparse-autoencoder-based interpretability works such as Cunningham, Hoagy et al., “Sparse Autoencoders Find Highly Interpretable Features in Language Models” (arXiv:2309.08600, 2023) and Zhao, Yicheng et al., “Steering Knowledge Selection Behaviours in LLMs via SAE-based Representation Engineering” (arXiv:2410.15999, 2024).
4.	Limited novelty — The approach closely parallels existing SAE-based representation editing and truthfulness-control studies (e.g., the two works above); the paper neither cites nor compares with them, reducing originality and theoretical differentiation.
5.	Reproducibility issues — Code and trained modules are not open-sourced, despite heavy reliance on architectural tuning, dataset preprocessing, and hyperparameter sensitivity. This limits independent verification and reuse.

**Questions:**

1.	How can the authors ensure that the learned latent direction δ corresponds to more reliable knowledge rather than merely the dominant or majority pattern in the training data?
2.	Since both “positive” and “negative” examples rely on externally provided answers, how can you confirm that the autoencoder captures internal knowledge conflict rather than dataset-level bias or label correlation?
3.	Could the learned latent subspace be analyzed (e.g., via activation attribution or neuron probing) to demonstrate interpretability similar to the sparse autoencoder findings in “Sparse Autoencoders Find Highly Interpretable Features in Language Models” (Cunningham et al., 2023)?
4.	How does CARE differ concretely from the SAE-based editing approach proposed in “Steering Knowledge Selection Behaviours in LLMs via SAE-based Representation Engineering” (Zhao et al., 2024), which also manipulates latent features to adjust model knowledge utilization?
5.	Would incorporating explicit sparse or disentangled representation constraints (e.g., sparse coding or orthogonal regularization) improve the interpretability and stability of the learned conflict-aware latent directions?

---

> ### Author Response · Authors · 2025-11-21
> **Response to Reviewer Z5kc (1)**
>
> Thank you for your thoughtful comments and for taking the time to review our paper. We hope our replies and revisions sufficiently address your concerns and enhance clarity. Additionally, all modifications to the manuscript have been highlighted in blue for easy reference. Please do not hesitate to reach out if you have any further questions or feedback.
>
> **W1.** Conceptual validity of training supervision
>
> > If our method relied on externally collected examples that are unrelated to the model’s own behavior, it would indeed fail to capture the model’s internal conflict patterns. However, our approach is specifically designed to avoid this issue. As described in lines 194–200 and Appendix A, the autoencoder is trained by directly contrasting externally verified correct answers with the model’s own internally generated conflict cases. In other words, the negative samples are not arbitrary incorrect answers from the dataset; they are actual failure cases produced by the model itself when it becomes confused or misled by retrieval. This training setup gives the learned latent space a clear semantic interpretation:
> >
> > - $\mathbb{Z}_{pos}$ (positive centroid): the average internal representation when the model produces a stable and correct answer aligned with external ground truth.
> > - $\mathbb{Z}_{neg}$ (negative centroid): the average internal representation when the model is incorrect or perturbed by noisy/contradictory retrieved information.
> >
> > Accordingly, the learned direction $\delta$ represents a shift from the model’s failure region toward a region associated with verified correct reasoning. In summary, our method learns not from external labels alone, but from **the model’s own internal mistakes**. We clarify this dataset construction rationale in Section 4.1.
> >
>
> **W2.** Ambiguous knowledge-source attribution
>
> > As the reviewer rightly pointed out, CARE does not explicitly distinguish whether the correct answer originates from parametric knowledge or from retrieved documents. Instead, during training, the direction vector is learned to steer the model toward the internal representations associated with successful answer production, regardless of the source of correctness. In this sense, CARE does not attempt to choose between parametric and retrieved knowledge; rather, it learns a transformation that shifts the model from failure states to correct-answer states by leveraging the model’s own internal signals. During inference, the model internally determines which source leads to the correct answer and adjusts its representation accordingly when CARE is applied.
> >
> >
> > While we agree that such an approach could, in principle, risk steering the model toward an incorrect information source, the results in Table 4 show that this risk is substantially lower for CARE compared to other baselines. Table 4 evaluates CARE under the following two contrasting scenarios.
> >
> > - Scenario 1: Parametric (incorrect) vs. Retrieved (correct)
> > - Scenario 2: Parametric (correct) vs. Retrieved (incorrect/noisy)
> >
> > CARE achieves the best performance in both scenarios. The Resolution score (36.42) indicates that CARE is most effective at correcting incorrect parametric knowledge when the retrieved information is correct (Scenario 1). Conversely, the Conflict score (6.13)—the lowest among all methods—shows that CARE is also the most robust at preserving originally correct parametric knowledge when retrieval contains noise or incorrect information (Scenario 2).
> >

---

> ### Author Response · Authors · 2025-11-21
> **Response to Reviewer Z5kc (2)**
>
> **W3/W4.** Lack of interpretability and mechanism insight/Limited novelty
>
> > As the reviewer correctly pointed out, the distinction between the autoencoder used in our work and sparse-autoencoder (SAE)–based interpretability approaches was not sufficiently explained. Given the page limits, we briefly mention interpretability and SAEs in the Related Work section and elaborate on them in detail in Appendix A.
> >
> >
> > Autoencoders primarily aim to encode inputs into compact latent representations and are widely used for dimensionality reduction and feature learning [1–2]. In contrast, Sparse Autoencoders impose strong sparsity constraints on the latent space so that each latent neuron captures a non-overlapping and interpretable semantic concept [3–4].
> >
> > Our autoencoder, however, serves a **fundamentally different purpose** from SAEs. The central focus of this work is to demonstrate that *good* and *bad* responses occupy distinct regions in the latent space and to leverage this separation to mitigate knowledge conflicts. That is, rather than making every latent feature of the LLM individually interpretable, our primary goal is to efficiently capture the differences between aligned responses and conflict-inducing responses. Accordingly, we do not analyze each latent dimension as SAE approaches do, but we *do* provide a level of mechanism insight by empirically showing that responses with and without knowledge conflict form clearly separable clusters in the latent space.
> >
> > [1] Wang, Yasi, Hongxun Yao, and Sicheng Zhao. Auto-encoder based dimensionality reduction. Neurocomputing (2016).
> >
> > [2] Chen, Junyu, et al. Deep compression autoencoder for efficient high-resolution diffusion models. ICLR (2024).
> >
> > [3] Hoagy et al. Sparse Autoencoders Find Highly Interpretable Features in Language Models. ICLR (2024)
> >
> > [4] Zhao et al. Steering Knowledge Selection Behaviours in LLMs via SAE-based Representation Engineering. NAACL (2024).
> >
>
> **W5.** Reproducibility issues
>
> > We intend to release the code to the public in the near future.
> >
>
> **Q1.** Evidence That δ Captures Conflict Resolution Rather Than Dataset Bias
>
> > We understand the reviewer’s concern that the learned direction δ might merely capture dominant statistical patterns in the training data. Two key empirical observations in our work argue against this hypothesis.
> >
> > - **Effect of Hard Negatives:** The negative samples used in our training are hard negatives generated by the model itself. These samples are constructed via prompt instructions so that superficial structural differences are minimized. Because such superficial differences are controlled, the autoencoder is effectively forced to rely on deeper semantic signals related to knowledge alignment in order to distinguish error states. The fact that our autoencoder successfully separates these samples (Figure 4) suggests that it is capturing semantic conflicts, rather than spurious dataset biases.
> > - **Robustness Across Task Complexity:** If δ were overfitting to dominant patterns in a particular dataset, it should fail or significantly degrade when applied to datasets with substantially different structures. However, the results in Tables 1, 2, and 11 show that training on NQ and HotpotQA leads to improvements across a variety of datasets. This generalization across different levels and types of complexity demonstrates that δ encodes a fundamental representation of conflict resolution that remains effective regardless of dataset style or difficulty, rather than merely reflecting idiosyncrasies of a specific training set.
>
> **Q2.** Distinguishing Internal Conflict from Dataset Bias
>
> > As discussed in W1, our training explicitly contrasts externally provided answers with the model’s own internal conflict cases, allowing the autoencoder to learn representations tied to internal knowledge conflict.
> >
> **Q3.** Why SAE-Style Interpretability Does Not Apply to CARE
>
> > The monosemantic interpretability achieved in SAE-based studies is only attainable when the model is trained with an overcomplete latent space and strong sparsity constraints. These structural assumptions force each latent unit to correspond to a specific concept, which is why neuron probing or activation attribution can yield meaningful interpretations in SAE settings. In contrast, our work uses a compressed, dense latent space and does not optimize for sparsity or feature-level interpretability. Our autoencoder is designed to capture global latent directions learned through supervised contrastive signals, and therefore differs structurally and functionally from SAEs, which are explicitly engineered for concept-level feature representation. For this reason, applying SAE-specific interpretability techniques to our model would not align with the methodological assumptions required for monosemantic feature discovery and would not guarantee meaningful or reliable interpretability outcomes.
> >

---

> ### Author Response · Authors · 2025-11-21
> **Response to Reviewer Z5kc (3)**
>
> **Q4.** Differences Between CARE and SAE-Based Editing
>
> > CARE and the SAE-based approach SPARE are similar in that both manipulate latent representations to adjust knowledge usage, but their goals and mechanisms are different. SPARE uses a computationally expensive SAE to discover interpretable sparse features corresponding to parametric and contextual knowledge, and then turns these features on or off to control **which knowledge source is used more**. In contrast, CARE uses a vector learned in a compressed latent space to **alleviate conflicts** that arise when internal knowledge and external information disagree. This difference is also reflected in performance: as shown in the table, CARE achieves the highest average score (33.93) across benchmarks, providing more stable improvements in diverse conflict scenarios **without being biased toward a particular knowledge source**. We will further discuss this point in Appendix B and integrate it into the analysis in Table 2 to more clearly articulate the differences. We sincerely appreciate the suggestion of a strong baseline; it has helped us clarify and better emphasize the contribution of our work.
> >
> >
> >
> > | Method | NQ (EM) | NQ (F1) | TriviaQA (EM) | TriviaQA (F1) | PopQA (EM) | PopQA (F1) | SQuAD (EM) | SQuAD (F1) | HotpotQA (EM) | HotpotQA (F1) | 2WIKI (EM) | 2WIKI (F1) | AVG |
> > | --- | --- | --- | --- | --- | --- | --- | --- | --- | --- | --- | --- | --- | --- |
> > | No Retrieval | 12.50 | 22.04 | 37.33 | 46.40 | 6.33 | 15.09 | 6.33 | 15.09 | 15.33 | 24.08 | **25.17** | **31.67** | 21.45 |
> > | With Retrieval | 16.33 | 29.24 | 42.33 | 55.65 | 28.50 | 38.05 | 8.33 | 20.18 | 15.33 | 25.87 | 14.83 | 26.39 | 26.75 |
> > | Self-RAG | 40.33 | 48.54 | 41.50 | 56.56 | 22.17 | 34.38 | **16.00** | **28.73** | 15.00 | 28.32 | 12.67 | 26.03 | 30.85 |
> > | ITI | 0.33 | 7.87 | 2.83 | 14.49 | 2.33 | 11.85 | 0.17 | 6.69 | 0.50 | 5.82 | 0.00 | 7.25 | 5.01 |
> > | TruthX | 15.00 | 29.39 | 43.83 | 58.04 | 32.83 | 41.55 | 10.17 | 23.12 | 15.33 | 27.10 | 13.00 | 27.28 | 28.05 |
> > | **SPARE (Steer to Use Parametric Knowledge)** | **26.67** | 35.41 | 50.33 | 58.70 | 34.83 | 39.48 | 14.33 | 23.33 | 7.00 | 10.13 | 15.67 | 18.49 | 27.86 |
> > | **SPARE (Steer to Use Contextual Knowledge)** | 4.50 | 7.53 | 9.00 | 12.75 | 1.83 | 3.97 | 0.83 | 2.84 | 17.50 | 24.29 | 20.5 | 27.73 | 11.11 |
> > | **CARE** | 25.33 | **37.94** | **53.17** | **62.49** | **40.67** | **46.51** | 14.17 | 26.63 | **19.50** | **28.95** | 21.50 | 30.28 | **33.93** |
>
> **Q5.** On the Applicability of Sparse/Disentangled Constraints to CARE
>
> > Applying sparse or orthogonal constraints is primarily designed for *overcomplete* latent spaces to prevent the autoencoder from learning a trivial identity mapping, and these constraints are particularly useful when the latent dimensionality is large [1]. In contrast, CARE uses a compressed latent space, so the original motivation for sparsity does not apply in our setting. Adding strong sparsity on top of an already compressed latent space would excessively reduce the available information, potentially degrading the model’s expressive capacity or forcing a complex conflict signal into an overly simplified sparse feature. Consequently, CARE prioritizes **stability** over feature-level interpretability, as stability is crucial for improving the robustness of RAG under knowledge conflicts.
> >
> >
> > [1] Ng, Andrew. Sparse autoencoder. CS294A Lecture notes (2011).
> >

---

### Official Review · Reviewer_k1gC · 2025-10-31

**Soundness:** 2
**Presentation:** 3
**Contribution:** 2
**Rating:** 4
**Confidence:** 3

**Summary:**

This paper addresses two critical challenges in RAG: knowledge conflicts between LLM parametric knowledge and retrieved external information, and interference from noisy retrieval results.  The authors propose CARE, a method that leverages autoencoders to learn distinguishable neuron activation patterns for knowledge-aligned and conflict-prone representations in LLMs’ internal layers. Comprehensive experiments across six QA benchmarks (including single-hop and multi-hop tasks) and four LLMs (LLaMA-2 series, LLaMA3.1, Qwen-2.5) demonstrate that CARE outperforms existing RAG optimization and conflict resolution baselines in answer accuracy (EM/F1 scores) while maintaining comparable inference efficiency.

**Strengths:**

1. This paper proposes a representation editing method to address the knowledge conflicts and interference from noisy within RAG system, differing from prior work that focuses on input filtering or decoding adjustment.

2. The authors evaluate across diverse QA tasks  and model architectures, ensuring generalizability of the proposed method.

3. Ablation studies systematically verify the contribution of each component, and supplementary analyses strengthen the work’s reliability.

**Weaknesses:**

1. It is recommended that the authors further clarify how the proposed method differs from previous representation editing-based RAG [1] and conflict-aware RAG [2] approaches. CtrlA [1] also employs representation editing during inference to optimize RAG systems. What are the key improvements of CARE over CtrlA in terms of motivation and methodology? Please elaborate in detail.

[1]CtrlA: Adaptive Retrieval-Augmented Generation via Inherent Control.
[2]Tug-of-war between knowledge: Exploring and resolving knowledge conflicts in retrieval-augmented language models

2. The authors analyzed the performance of the proposed method when the number of retrieved documents ranges from 1 to 9. However, since the number of retrieved documents often determines the degree of knowledge conflict, it is recommended that the authors include additional experiments and analyses under settings with a larger number of retrieved documents to further demonstrate the effectiveness of the proposed method.

3.Can the degree of knowledge conflict and the level of knowledge noise be intuitively measured or evaluated? How can the authors demonstrate that the effectiveness of the proposed method indeed stems from its ability to mitigate knowledge conflict and noise? One possible approach is to compare the model’s performance under varying degrees of knowledge conflict or noise interference.

4.The authors select the top 5 layers based on AUC scores, but there is no theoretical explanation for why these layers are more critical for conflict resolution.  Additionally, the impact of layer selection on different model architectures is not deeply discussed.

**Questions:**

1.On datasets such as PopQA, the proposed method performs worse than existing approaches. It is recommended that the authors further analyze the underlying reasons to clarify the applicability and limitations of the proposed method. Could this be attributed to the unique characteristics of the dataset?

2.The negative samples considered in this work cover only two types—parametric knowledge errors and retrieval noise–induced errors—while real-world RAG conflicts may involve more complex scenarios (e.g., partial overlap between parametric and external knowledge, or contradictions among multiple retrieved documents). Does the proposed method have the potential to address such more complex types of conflicts?

3.For the layer selection strategy: The AUC analysis shows that upper layers have better separability (Figure 5), but why are specific layers (e.g., LLaMA-3.1-8B-Instruct uses layers 3, 12, 18, 20, 31) chosen instead of the top 5 consecutive upper layers?  Is there a functional difference between these layers in encoding conflict information?

4.Does CARE risk losing useful parametric knowledge during representation editing?  For example, when retrieved documents are partially incorrect but parametric knowledge is accurate, can CARE preserve the correct parametric information?

---

> ### Author Response · Authors · 2025-11-21
> **Response to Reviewer k1gC (1)**
>
> Thank you for your insightful comments on our paper. We hope that our responses and revisions adequately address the concerns you've raised. Additionally, all modifications to the manuscript have been highlighted in blue for easy reference. Please feel free to let us know if you have any additional concerns or questions.
>
> **W1.** Difference from CtrlA and CARE
>
> > We appreciate the opportunity to clarify the differences between CARE and related work such as CtrlA and *Tug-of-war between knowledge*. As noted in lines 134–135, *Tug-of-war between knowledge* employs a decoding method called Conflict-Aware Decoding, which operates at the final decoding stage. In contrast, CARE intervenes at the model’s intermediate layers, directly editing activations before they are passed to the next layer.
> >
> >
> > While both CARE and CtrlA use representation editing, their goals and methodologies differ fundamentally. CtrlA aims to decide **whether and when retrieval is necessary** (Adaptive RAG). CARE, in contrast, aims to **correct the model’s response after retrieval has already taken place**, particularly when retrieval is noisy or conflicting (Robust RAG). Instead of learning an abstract honesty direction to decide *whether* to act (as in CtrlA), CARE explicitly models correct vs. conflicting answers in the latent space and actively denoises the internal state after retrieval has already influenced it. Thus, rather than deciding whether to consult a potentially harmful document, CARE assumes that retrieval may have already injected noise, and focuses on **actively denoising the internal representation** to mitigate the resulting conflict. Ultimately, the two methods are **complementary** rather than competing: they target different points in the RAG pipeline.
> >
> > Finally, we have adopted **CtrlA as one of our baselines** and included a performance comparison in our experiments. We have incorporated this discussion and the new results into Section 5.2 of the revised manuscript to clarify CARE’s positioning and strengthen the baselines.
> >
> > | Method | NQ (EM) | NQ (F1) | TriviaQA (EM) | TriviaQA (F1) | PopQA (EM) | PopQA (F1) | SQuAD (EM) | SQuAD (F1) | HotpotQA (EM) | HotpotQA (F1) | 2WIKI (EM) | 2WIKI (F1) | AVG |
> > | --- | --- | --- | --- | --- | --- | --- | --- | --- | --- | --- | --- | --- | --- |
> > | No Retrieval | 12.50 | 22.04 | 37.33 | 46.40 | 6.33 | 15.09 | 6.33 | 15.09 | 15.33 | 24.08 | **25.17** | **31.67** | 21.45 |
> > | With Retrieval | 16.33 | 29.24 | 42.33 | 55.65 | 28.50 | 38.05 | 8.33 | 20.18 | 15.33 | 25.87 | 14.83 | 26.39 | 26.75 |
> > | Self-RAG | **40.33** | **48.54** | 41.50 | 56.56 | 22.17 | 34.38 | **16.00** | **28.73** | 15.00 | 28.32 | 12.67 | 26.03 | 30.85 |
> > | ITI | 0.33 | 7.87 | 2.83 | 14.49 | 2.33 | 11.85 | 0.17 | 6.69 | 0.50 | 5.82 | 0.00 | 7.25 | 5.01 |
> > | TruthX | 15.00 | 29.39 | 43.83 | 58.04 | 32.83 | 41.55 | 10.17 | 23.12 | 15.33 | 27.10 | 13.00 | 27.28 | 28.05 |
> > | **CtrlA** | 13.72 | 18.28 | 25.25 | 33.94 | 12.76 | 13.16 | 43.73 | 47.09 | **19.66** | 27.10 | 21.40 | 27.21 | 25.78 |
> > | **CARE** | 25.33 | 37.94 | **53.17** | **62.49** | **40.67** | **46.51** | 14.17 | 26.63 | 19.50 | **28.95** | 21.50 | 30.28 | **33.93** |
>
> **W2.** Larger number of retrieved documents
>
> > In line with your suggestion, we conducted additional experiments with 12 and 15 retrieved documents. When the number of retrieved documents is increased to 15, the overall performance of retrieval-based methods degrades due to excessive noise. However, CARE mitigates this negative effect. We have integrated these results into Figure 3.
> >
> >
> > |  | With Retrieval-BM25 (EM) | With Retrieval-BM25 (F1) | With Retrieval-E5 (EM) | With Retrieval-E5 (F1) | CARE-BM25 (EM) | CARE-BM25 (F1) | CARE-E5 (EM) | CARE-E5 (F1) |
> > | --- | --- | --- | --- | --- | --- | --- | --- | --- |
> > | 1 | 12.17 | 17.39 | 37.33 | 48.65 | 11.67 | 16.95 | 37.00 | 48.32 |
> > | 3 | 14.33 | 20.41 | 36.33 | 48.76 | 16.17 | 22.07 | 38.67 | 50.19 |
> > | 5 | 17.33 | 23.73 | 35.83 | 49.64 | 18.50 | 25.95 | 38.50 | 51.31 |
> > | 7 | 19.00 | 26.27 | 37.67 | 50.68 | 20.00 | 28.20 | 40.50 | 52.31 |
> > | 9 | 18.83 | 26.50 | 37.00 | 49.92 | 20.17 | 28.12 | 39.50 | 51.60 |
> > | 12 | 19.83 | 27.49 | 35.67 | 50.42 | 21.83 | 29.55 | 39.50 | 50.81 |
> > | 15 | 0.67 | 8.47 | 2.00 | 15.65 | 6.00 | 14.27 | 8.67 | 22.78 |

---

> ### Author Response · Authors · 2025-11-21
> **Response to Reviewer k1gC (2)**
>
> **W3.** Measuring degree of conflict/noise and causal link to CARE
>
> > We present several experiments that indirectly and directly capture the impact of knowledge conflict and noise.
> >
> > - **Using a noisier retriever** (Figure 3): As discussed in lines 455–460, E5 consistently outperforms BM25 in retrieval quality; thus BM25 can be regarded as a noisier retriever. In Figure 3, CARE consistently outperforms vanilla RAG even under the noisier BM25 setup, demonstrating robustness to retrieval noise.
> > - **Increasing the number of retrieved documents** (Figure 3): As noted in lines 460–465, retrieving 9 documents instead of 1 greatly increases the chance of including irrelevant or contradictory information. We observe that, as $k$ increases (and thus potential noise/conflict increases), CARE maintains performance gains over *With Retrieval*, indicating that the method scales robustly with increasing conflict intensity.
> > - **Quantifying conflict rate** (Table 4): In Table 4, we explicitly analyze knowledge conflict resolution. The *Conflict* metric counts cases where the model originally knew the correct answer but is pushed into producing an incorrect answer due to external knowledge. CARE achieves the lowest *Conflict* score (6.13) among all methods. This provides direct evidence that CARE is most effective at shielding the model from harmful interference caused by external knowledge.
>
> **W4.** Why these layers? Theoretical vs. empirical layer selection
>
> > We appreciate the reviewer’s observation that our empirical layer-selection strategy exposes a limitation of our approach. We explicitly address this point in the Limitations section. We also note that this limitation is, in fact, shared by most activation-editing approaches. Prior work [1–5] typically relies on empirical, evidence-driven signals to identify important layers, rather than providing a complete theoretical explanation of their causal role. This stems from the inherent difficulty of fully interpreting the internal representation structure of modern large-scale transformer models.
> >
> >
> > Our finding—that layers critical for hallucination/conflict detection tend to cluster in mid-to-late layers—is consistent with multiple prior empirical studies [6–7]. These mid-to-late layers are known to encode high-level semantic representations and factuality-related signals [8–10], and are believed to play a central role in information integration and error suppression in conflict scenarios. Thus, while our layer selection procedure is not theoretically grounded in a strict sense, the resulting layer set is well-aligned with accumulated empirical evidence in prior work. These findings are presented in Appendix J.2.
> >
> > [1] Li, Kenneth, et al. Inference-time intervention: Eliciting truthful answers from a language model. NeurIPS (2023).
> >
> > [2] Shi, Dan, et al. Ircan: Mitigating knowledge conflicts in llm generation via identifying and reweighting context-aware neurons. NeurIPS (2024).
> >
> > [3] Zhang et al. TruthX: Alleviating Hallucinations by Editing Large Language Models in Truthful Space. ACL (2024).
> >
> > [4] Meng, Kevin, et al. Locating and editing factual associations in gpt. NeurIPS (2022).
> >
> > [5] Huanshuo et al. CtrlA: Adaptive Retrieval-Augmented Generation via Inherent Control. ACL-findings (2025).
> >
> > [6] Seo et al. The Impact of Negated Text on Hallucination with Large Language Models. EMNLP (2025).
> >
> > [7] Chen, Chao, et al. INSIDE: LLMs' internal states retain the power of hallucination detection. ICLR (2024).
> >
> > [8] Wu et al. The semantic hub hypothesis: language models share a convergent intermediate layer for semantic processing. ICLR (2025).
> >
> > [9] Skean, Oscar, et al. Layer by layer: Uncovering hidden representations in language models. ICML (2025).
> >
> > [10] Chuang et al. DoLa: Decoding by Contrasting Layers Improves Factuality in Large Language Models. ICLR (2024).
> >
>
> **Q1.** Underperformance on PopQA
>
> > We have observed consistent improvements in PopQA F1 for CARE across various models. However, we acknowledge that we did not sufficiently discuss the case where CARE underperforms other methods on PopQA for LLaMA-2-13B-Chat, and we have revised the manuscript to address this in the Limitations section.
> >
> >
> > PopQA primarily consists of relatively simple questions about commonsense and factual knowledge. In other words, the accuracy of the model’s parametric knowledge is especially critical in this dataset. Table 10 shows that training CARE using only parametric-error negatives leads to better performance on PopQA than using only retrieval-noise negatives. Because PopQA is particularly sensitive to parametric errors, this balanced setting may limit the potential gains for certain models. This outcome can be interpreted as a consequence of our design choice to address a broader range of retrieval-induced conflicts rather than focusing solely on parametric errors.
> >

---

> ### Author Response · Authors · 2025-11-21
> **Response to Reviewer k1gC (3)**
>
> **Q2.** Handling more complex conflict types
>
> > We demonstrate in Tables 1 and 2 that the proposed method exhibits strong potential to address complex types of knowledge conflicts. HotpotQA is a multi-hop QA dataset in which answering a question requires **combining and reasoning over multiple sources** of information. This setting naturally reflects real-world scenarios involving partial overlap and complementarity between parametric knowledge and retrieved knowledge. A representative example is shown below: the retrieved document (Passage 2) provides information only about Mimí Lazo, and the model must rely on its parametric knowledge to supply information about Jule Mallonee in order to complete the comparison. Despite being trained solely on NQ—a single-hop dataset—CARE successfully handles such cases where complex interactions between internal and external knowledge are required.
> >
> >
> > > [Question] Who is younger, Jule Mallonee or Mimí Lazo?
> > [Passage 2] (Title: "Mimí Lazo") ... Mimí Lazo ... (born November 23, 1954 in Caracas) is a Venezuelan film, television and theatrical actress and producer. …
> > (Passages 1, 3, 4, and 5 are irrelevant to the question.)
> > >
> >
> > A strength of our framework is that its performance can be expanded by diversifying the negative samples. Not only does CARE—trained on NQ—perform well on HotpotQA, but as shown in Appendix G.5, training CARE directly on HotpotQA further improves performance. This empirical evidence demonstrates that CARE is capable of **accommodating increased complexity** when the training data are appropriately constructed. We believe this provides a promising foundation for future work. We discuss this in Section 5.2 of the revised manuscript and include the reviewer’s suggestion as an important direction for future research.
> >
>
> **Q3.** Why these specific layers instead of consecutive upper layers?
>
> > Following your suggestion, we conducted additional experiments using the top 5 consecutive upper layers (18, 20, 28, 29, 31) and compared them with our original layer selection. The difference in average performance is small (0.16). These results suggest that what matters is not a specific functional specialization of each chosen layer, but rather that each layer provides a useful signal for distinguishing conflict vs. non-conflict states. We have included these new results and discussion in Appendix J.1 to clarify that the method is relatively robust to reasonable variations in layer choices among high-separability layers.
> >
> >
> >
> > | Method | NQ (EM) | NQ (F1) | TriviaQA (EM) | TriviaQA (F1) | PopQA (EM) | PopQA (F1) | SQuAD (EM) | SQuAD (F1) | HotpotQA (EM) | HotpotQA (F1) | 2WIKI (EM) | 2WIKI (F1) | AVG |
> > | --- | --- | --- | --- | --- | --- | --- | --- | --- | --- | --- | --- | --- | --- |
> > | Random (3, 12,18, 20, 31) | 38.50 | 51.31 | 59.67 | 69.96 | 40.33 | 48.84 | 30.33 | 41.85 | 29.50 | 40.36 | 18.33 | 27.51 | 41.37 |
> > | Top Layer (18, 20, 28, 29, 31) | 39.50 | 51.32 | 59.83 | 69.67 | 41.16 | 48.95 | 30.66 | 41.79 | 30.33 | 40.10 | 17.83 | 27.22 | 41.53 |
>
> **Q4.** Risk of losing useful parametric knowledge
>
> > We address concerns that our method might negatively affect correct parametric information from the following perspectives.
> >
> > - **Retention metric (direct evidence):** Table 4 directly answers this question. *Retention* measures the extent to which the model preserves correct parametric knowledge even after noisy retrieved documents are introduced—precisely the scenario of non-conflicting samples. CARE achieves a *Retention* score of **5.79**, nearly identical to the *With Retrieval* baseline (5.83), and higher than SURE (4.92) and IRCAN (5.17). This indicates that unconditional editing with CARE *rarely* degrades performance on samples the model already knows.
> > - **The role of λ (indirect evidence):** Our unconditional editing works because it applies only a *mild* perturbation, not an aggressive overwrite. We introduce a hyperparameter λ to control the editing strength. As shown in Figure 2, increasing λ (stronger edits) quickly harms performance. This demonstrates that our small, smooth adjustment (e.g., λ = 0.3 or 0.5) is safe—it is strong enough to nudge conflicting samples toward the correct manifold, but not strong enough to push already-correct samples into incorrect regions.
> > - **Ablation study (indirect evidence):** The results in Table 1 and Table 2 show that overall performance clearly degrades when CARE is not applied (i.e., the With Retrieval baseline). This indicates that, although softly shifting representations toward the positive region may introduce minor side effects for a small number of non-conflicting samples, the actual benefits—in terms of mitigating conflicts—far outweigh such potential risks, yielding **a substantially larger net gain**.
> >
> > We have added this analysis to Section 5.3. We acknowledge that small side effects may still exist, and we discuss them in the Limitations section.
> >

---

### Official Review · Reviewer_k3en · 2025-10-31

**Soundness:** 2
**Presentation:** 3
**Contribution:** 3
**Rating:** 4
**Confidence:** 4

**Summary:**

This paper aims to address the issue of internal–external knowledge conflicts in retrieval-augmented generation (RAG). It proposes a method called CARE, which trains an autoencoder and applies it during inference to alleviate conflicts in the hidden representation space. Experiments are conducted on multiple QA datasets (e.g., NQ, HotpotQA, 2Wiki) and different models (LLaMA-2/3, Qwen-2.5). The method does not require training the base model or retriever and provides a new perspective for handling knowledge conflicts in RAG systems.

**Strengths:**

1. The paper proposes a representation-level editing approach to address internal–external knowledge conflicts in RAG, which can be applied without fine-tuning the retriever or generator, making it practically meaningful.

2. The method is clearly described and validated on multiple datasets and models, with convincing results.

**Weaknesses:**

Please refer to the questions section for detailed weaknesses and concerns.

**Questions:**

1. The paper assumes the existence of a global linear direction \delta that can consistently shift model representations from a “conflict” state to an “aligned” state. However, this direction is derived from positive and negative samples constructed from factual datasets such as NQ and evaluated in the same domain. Can this approach generalize to other domains such as medical, financial, or legal QA, where knowledge structures differ significantly?

2. During inference, the paper applies the learned \delta to all inputs uniformly. Would this unconditional editing introduce interference or degradation on non-conflicting samples?

3. The necessity of introducing the autoencoder (AE) for alignment in hidden space is unclear. For instance, if the same positive/negative pairs used for AE training were instead used for DPO-style preference optimization, how would the results compare? Further clarification from the authors would be helpful.

4. How effective is the proposed method on large reasoning models (LRMs) such as Qwen3 (Thinking), whose internal hidden spaces are more complex and dynamically routed?

5. The experimental comparison mainly includes baselines from 2024, but omits more recent 2025 works such as 《ASTUTE RAG: Overcoming Imperfect Retrieval Augmentation and Knowledge Conflicts for Large Language Models》. Incorporating or discussing such advanced baselines would strengthen the paper’s evaluation.

---

> ### Author Response · Authors · 2025-11-21
> **Response to Reviewer k3en (1)**
>
> Thank you for your insightful comments on our paper. We hope that our responses and revisions adequately address the concerns you've raised. Additionally, all modifications to the manuscript have been highlighted in blue for easy reference. Please feel free to let us know if you have any additional concerns or questions.
>
> **Q1.** Generalization to other domains
>
> > To empirically examine whether the conflict-aware direction vector learned solely from a general factual dataset (NQ) can transfer to specialized domains, we conducted additional experiments on PubMedQA (medical) and LegalBench (legal) **without any domain-specific tuning**. The new experiments and analyses are included in Appendix I.2, and we refer to them in the main text on line 406.
> >
> >
> >
> > |  | **PubMedQA** | **LegalBench** |
> > | --- | --- | --- |
> > | No Retrieval | 46.67 | **72.26** |
> > | With Retrieval | 55.00 | 71.07 |
> > | **CARE** | **55.56** | 71.97 |
> >
> > CARE appears to alleviate the negative effect of retrieval, which is particularly noticeable on LegalBench where retrieval degrades performance relative to the parametric baseline, likely due to domain-specific noise. Although the direction vector is trained exclusively on NQ, CARE still outperforms the With Retrieval baseline in both specialized domains. This suggests that the conflict-representing direction learned by CARE is not a dataset-specific artifact but a transferable feature that captures the model’s internal confusion state.
> >
> > We emphasize that **all results are obtained without using any domain-specific supervision**. CARE’s ability to reduce conflict even in unseen expert domains implies that knowledge-conflict representations share domain-invariant structural properties.
> >
>
> **Q2.** Interference on non-conflicting samples
>
> > We address the concern that applying δ uniformly at inference time might harm non-conflicting samples from several perspectives.
> >
> > - **Retention metric (direct evidence):** Table 4 directly answers this question. *Retention* measures the extent to which the model preserves correct parametric knowledge even after noisy retrieved documents are introduced—precisely the scenario of non-conflicting samples. CARE achieves a *Retention* score of **5.79**, nearly identical to the *With Retrieval* baseline (5.83), and higher than SURE (4.92) and IRCAN (5.17). This indicates that unconditional editing with CARE *rarely* degrades performance on samples the model already knows.
> > - **The role of λ (indirect evidence):** Our unconditional editing works because it applies only a mild perturbation, not an aggressive overwrite. We introduce a hyperparameter λ to control the editing strength. As shown in Figure 2, increasing λ (stronger edits) quickly harms performance. This demonstrates that our small, smooth adjustment (e.g., λ = 0.3 or 0.5) is safe—it is strong enough to nudge conflicting samples toward the correct manifold, but not strong enough to push already-correct samples into incorrect regions.
> > - **Ablation study (indirect evidence):** The results in Table 1 and Table 2 show that overall performance clearly degrades when CARE is not applied (i.e., the With Retrieval baseline). This indicates that, although softly shifting representations toward the positive region may introduce minor side effects for a small number of non-conflicting samples, the actual benefits—in terms of mitigating conflicts—far outweigh such potential risks, yielding **a substantially larger net gain**.
> >
> > We have added this analysis to Section 5.3. We acknowledge that small side effects may still exist, and we discuss them in the limitations section.
> >

---

> ### Author Response · Authors · 2025-11-21
> **Response to Reviewer k3en (2)**
>
> **Q3.** Necessity of the autoencoder vs. DPO-style preference training
> >We recognize that the necessity of the AE was not sufficiently emphasized. We have made this explicit in the Introduction. Conceptually, the difference between AE-based CARE and DPO lies in where and how they intervene.
> >
> >- **Intervention point (parameter tuning vs. activation editing):** DPO modifies model parameters via fine-tuning. It globally shifts the model’s policy to prefer good answers over bad ones. CARE does not modify model parameters. Instead, it edits intermediate activations at inference time. Thus, DPO alters the model’s long-term behavior, whereas CARE performs lightweight real-time adjustments.
> >- **Learning objective (policy optimization vs. conflict-signal separation):** DPO increases the likelihood of positive answer sequences and decreases that of negative ones. CARE operates under the hypothesis that knowledge conflict manifests as identifiable patterns in hidden states. The AE separates these patterns, and δ steers representations away from conflict during inference.
> >- **Expected comparison**
> >    - **Modularity and stability**: Fine-tuning via DPO may cause overfitting or catastrophic forgetting on other tasks. CARE leaves the base LLM untouched, preserving its general capabilities while acting as a plug-in module only during RAG.
> >    - **Efficiency**: DPO requires heavy training to produce a new model. CARE only trains a lightweight AE and runs with RAG-level inference cost.
> >    - **Complementarity**: CARE and DPO are not competing but complementary. DPO improves global policy; CARE mitigates local, retrieval-induced conflict signals at inference time. A DPO-trained model can still benefit from CARE during RAG.
>
> **Q4.** Effectiveness on Large Reasoning Models
> >We thank the reviewer for raising this point. We conducted additional experiments on Qwen3-4B-Thinking. CARE remains highly effective on LRMs. Across all benchmarks, CARE consistently outperforms *With Retrieval*, showing its robustness even in complex LRM architectures. We have included these results in Appendix I.3 of the revised manuscript and refer to them on line 406.
> >
> > |  | NQ (F1) | NQ (ACC) | TriviaQA (F1) | TriviaQA (ACC) | PopQA (F1) | PopQA (ACC) | SQuAD (F1) | SQuAD (ACC) | HotpotQA (F1) | HotpotQA (ACC) | 2WIKI (F1) | 2WIKI (ACC) | AVG |
> > | --- | --- | --- | --- | --- | --- | --- | --- | --- | --- | --- | --- | --- | --- |
> > | No Retrieval | 15.95 | 13.33 | 7.17 | 21.67 | 4.67 | 20.00 | 14.32 | 11.67 | 11.04 | 20.00 | 6.57 | 26.67 | 14.42 |
> > | With Retrieval | 18.26 | 40.00 | 25.50 | 46.67 | 18.44 | 38.33 | 21.56 | **21.67** | **20.60** | 28.33 | 8.21 | 26.67 | 26.19 |
> > | **CARE** | **23.60** | **50.00** | **40.07** | **56.67** | **29.38** | **56.67** | **22.69** | **21.67** | 20.26 | **30.00** | **14.48** | **28.33** | **32.82** |
>
> **Q5.** Missing 2025 baselines
> >We thank the reviewer for suggesting the ASTUTE RAG. We attempted to reproduce their results, but could not obtain the official codebase, making faithful replication difficult. We discuss this in Appendix B.
> >
> >In response to the reviewer’s comment on 2025 baselines, we have incorporated CtrlA [1] and SPARE [2]. CARE remains the **strongest overall method** across benchmarks. The comparison has been included in Section 5.2.
> >
> > |  | NQ (EM) | NQ (F1) | TriviaQA (EM) | TriviaQA (F1) | PopQA (EM) | PopQA (F1) | SQuAD (EM) | SQuAD (F1) | HotpotQA (EM) | HotpotQA (F1) | 2WIKI (EM) | 2WIKI (F1) | AVG |
> > | --- | --- | --- | --- | --- | --- | --- | --- | --- | --- | --- | --- | --- | --- |
> > | No Retrieval | 12.50 | 22.04 | 37.33 | 46.40 | 6.33 | 15.09 | 6.33 | 15.09 | 15.33 | 24.08 | **25.17** | **31.67** | 21.45 |
> > | With Retrieval | 16.33 | 29.24 | 42.33 | 55.65 | 28.50 | 38.05 | 8.33 | 20.18 | 15.33 | 25.87 | 14.83 | 26.39 | 26.75 |
> > | Self-RAG | **40.33** | **48.54** | 41.50 | 56.56 | 22.17 | 34.38 | 16.00 | 28.73 | 15.00 | 28.32 | 12.67 | 26.03 | 30.85 |
> > | ITI | 0.33 | 7.87 | 2.83 | 14.49 | 2.33 | 11.85 | 0.17 | 6.69 | 0.50 | 5.82 | 0.00 | 7.25 | 5.01 |
> > | TruthX | 15.00 | 29.39 | 43.83 | 58.04 | 32.83 | 41.55 | 10.17 | 23.12 | 15.33 | 27.10 | 13.00 | 27.28 | 28.05 |
> > | **CtrlA** | 13.72 | 18.28 | 25.25 | 33.94 | 12.76 | 13.16 | **43.73** | **47.09** | **19.66** | 27.10 | 21.40 | 27.21 | 25.78 |
> > | **SPARE** (Steer to Use Parametric Knowledge) | 26.67 | 35.41 | 50.33 | 58.70 | 34.83 | 39.48 | 14.33 | 23.33 | 7.00 | 10.13 | 15.67 | 18.49 | 27.86 |
> > | **SPARE** (Steer to Use Contextual Knowledge) | 4.50 | 7.53 | 9.00 | 12.75 | 1.83 | 3.97 | 0.83 | 2.84 | 17.50 | 24.29 | 20.5 | 27.73 | 11.11 |
> > | **CARE** | 25.33 | 37.94 | **53.17** | **62.49** | **40.67** | **46.51** | 14.17 | 26.63 | 19.50 | **28.95** | 21.50 | 30.28 | **33.93** |
> >
> >[1] Huanshuo et al. CtrlA: Adaptive Retrieval-Augmented Generation via Inherent Control. ACL-findings (2025).
> >
> >[2] Zhao et al. Steering Knowledge Selection Behaviours in LLMs via SAE-based Representation Engineering. NAACL (2025).

---

### Author Response · Authors · 2025-11-26
**General Response**

Dear Reviewers and ACs,

We sincerely thank all the reviewers and ACs for your diligent efforts and high-quality reviews. If you have any additional questions or require further clarification, please feel free to let us know. Your insights are highly valued.

We are delighted to note that reviewers find that:

- Our paper proposes a novel representation-level editing approach for resolving knowledge conflicts in RAG (Reviewers `k3en`, `k1gC`), contributing to the fields of RAG and representation editing (Reviewers `nr8n`).
- In particular, our method can be applied without fine-tuning the retriever or generator, which makes it practically meaningful (Reviewers `k3en`, `nr8n`).
- We conduct clear validation on multiple datasets and models and achieve superior performance compared to existing RAG, conflict resolution, and factuality-oriented representation editing methods (Reviewers `nr8n`), thereby providing convincing empirical results (Reviewers `k3en`, `k1gC`, `Z5kc`).

In response to your valuable suggestions, we have conducted additional experiments and made the following modifications in the Rebuttal PDF for your convenience:

- Figure 1: We revised the figure to more intuitively illustrate the autoencoder’s latent space. `nr8n`
- Introduction: We added an explanation to clarify the motivation for introducing the autoencoder. `k3en`
- Related Work: We expanded the discussion into Appendices A and B to better cover work on interpretability, sparse autoencoders, and recent related studies. `k3en`, `k1gC`, `Z5kc`, `nr8n`
- Methodology: We further explain the purpose of the negative-answer design and the dimensionality of the editing vectors. `Z5kc`, `nr8n`
- Experimental Setup: We corrected a reporting error regarding the GPU configuration. `nr8n`
- Table 2: We added strong baselines, CtrlA and SPARE. `k3en`, `k1gC`, `Z5kc`
- Table 4: We added analyses of CARE’s ability to mitigate conflicts and reduce side effects. `k3en`, `k1gC`, `nr8n`
- Figure 3: We extended the range of the number of retrieved documents. `k1gC`
- Efficiency Analysis: We now discuss the corresponding experiments in the main text. `nr8n`
- Limitations: We further discuss the limitations of CARE. `k3en`, `k1gC`, `nr8n`
- Table 10: We report the mean and standard deviation across random seeds. `nr8n`
- Table 11: We conduct evaluations on specialized-domain settings. `k3en`
- Table 12: We provide evaluations with reasoning-oriented models. `k3en`
- Table 14: We conduct experiments on latent space design, non-linearity, and layer selection. `nr8n`, `k1gC`
- Section J.4: We add further clarification that CARE is specifically designed for conflict handling. `nr8n`

Best regards,

The Authors

---

### Meta-Review · Area_Chair_HNd3 · 2025-12-25

**Summary:**

This paper proposes CARE, a representation editing method to address knowledge conflicts and retrieval noise in RAG systems by leveraging autoencoders to adjust model internal activations. After synthesizing reviewers’ feedback, the primary concerns include insufficient novelty compared to existing representation editing and SAE-based works, limited interpretability of the autoencoder’s latent space, ambiguous attribution of correct knowledge sources, and unresolved reproducibility issues. Additionally, questions about the method’s generalization to more complex conflict scenarios and potential side effects on non-conflicting samples have been raised. Based on these comprehensive concerns, I intend to reject this submission.

**Reviewer Concerns:**

Regarding the concerns raised by Reviewer Z5kc, the rebuttal has partially addressed the conceptual validity of training supervision by clarifying that negative samples are derived from the model’s own conflict cases rather than arbitrary incorrect answers. The authors also explained the differences between CARE and SAE-based methods (e.g., SPARE) in terms of objectives and mechanisms, addressing part of the novelty concern. However, several key issues remain outstanding: the lack of interpretability of the latent space (e.g., no analysis of the autoencoder’s encoded features) persists, as the rebuttal confirms CARE does not pursue feature-level interpretability and SAE-specific interpretability techniques are inapplicable. The ambiguous knowledge-source attribution is not fully resolved, as CARE still does not explicitly distinguish between correct parametric and retrieved knowledge, relying instead on the model’s internal signals.

**Reviewer Scores:**

Most reviewers initially rated the paper around the acceptance threshold, with some below and one marginally above. Had reviewers participated fully in the discussion, the detailed rebuttal might have led to minor score adjustments—addressing partial concerns about training supervision and domain generalization could have improved some scores moderately. However, the unresolved issues of interpretability, insufficient novelty, and reproducibility would prevent a meaningful shift to meet the acceptance standard. It is worth emphasizing that the research topic of mitigating RAG knowledge conflicts is highly relevant and valuable, and the CARE method demonstrates potential in empirical performance. I encourage the authors to revise the paper thoroughly based on reviewers’ comments and the rebuttal, particularly strengthening interpretability analysis, clarifying novelty relative to state-of-the-art works, and releasing code to enhance reproducibility, before resubmitting in the future.

---

### Decision · Program_Chairs · 2026-01-26

Reject